
# Impact of molecular structure on secondary organic aerosol formation from aromatic hydrocarbon photooxidation under low NO$_x$ conditions

**L. Li[1,2], P. Tang[1,2], S. Nakao[1,2,a], and D. R. Cocker III[1,2]**

[1]University of California, Riverside, Department of Chemical and Environmental Engineering, Riverside, CA 92507, USA
[2]College of Engineering-Center for Environmental Research and Technology (CE-CERT), Riverside, CA 92507, USA
[a]currently at: Clarkson University, Department of Chemical and Biomolecular Engineering, Potsdam, NY 13699, USA

Received: 24 October 2015 – Accepted: 26 November 2015 – Published: 15 January 2016

Correspondence to: D. R. Cocker III (dcocker@engr.ucr.edu)

Published by Copernicus Publications on behalf of the European Geosciences Union.

Discussion Paper | Discussion Paper | Discussion Paper | Discussion Paper

**ACPD**

doi:10.5194/acp-2015-871

**Impact of molecular structure on SOA formation**

L. Li et al.

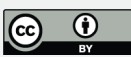

## Abstract

The molecular structure of volatile organic compounds (VOC) determines their oxidation pathway, directly impacting secondary organic aerosol (SOA) formation. This study comprehensively investigates the impact of molecular structure on SOA formation from the photooxidation of twelve different eight to nine carbon aromatic hydrocarbons under low $NO_x$ conditions. The effects of the alkyl substitute number, location, carbon chain length and branching structure on the photooxidation of aromatic hydrocarbons are demonstrated by analyzing SOA yield, chemical composition and physical properties. Aromatic hydrocarbons, categorized into five groups, show a yield order of ortho (*o*-xylene and *o*-ethyltoluene) > one substitute (ethylbenzene, propylbenzene and isopropylbenzene) > meta (*m*-xylene and *m*-ethyltoluene) > three substitute (trimethylbenzenes) > para (*p*-xylene and *p*-ethyltoluene). SOA yields of aromatic hydrocarbon photooxidation do not monotonically decrease when increasing alkyl substitute number. The ortho position promotes SOA formation while the para position suppresses aromatic oxidation and SOA formation. Observed SOA chemical composition and volatility confirm that higher yield is associated with further oxidation. SOA chemical composition also suggests that aromatic oxidation increases with increasing alkyl substitute chain length and branching structure. Further, carbon dilution theory developed by Li et al. (2015a) is extended in this study to serve as a standard method to determine the extent of oxidation of an alkyl substituted aromatic hydrocarbon.

## 1   Introduction

Organic aerosols are critical to human health (Dockery et al., 1993; Krewski et al., 2003; Davidson et al., 2005), climate change (IPPC, 2007) and visibility (Pöschl, 2005; Seinfeld and Pandis, 2006). Global anthropogenic secondary organic aerosol (SOA) sources are underestimated by current models (Henze et al., 2008; Matsui et al., 2009; Hallquist et al., 2009; Farina et al., 2010) and have larger growth potential than biogenic

Discussion Paper | Discussion Paper | Discussion Paper | Discussion Paper |

**ACPD**

doi:10.5194/acp-2015-871

**Impact of molecular structure on SOA formation**

L. Li et al.

aerosol sources due to the increase of known anthropogenic emissions (Heald et al., 2008). Therefore, it is crucial to explore SOA formation mechanism from anthropogenic precursors.

Aromatic hydrocarbons are major anthropogenic SOA precursors (Kanakidou et al., 2005; Henze et al., 2008; Derwent et al., 2010). $C_8$ (ethylbenzene, xylenes) and $C_9$ (ethyltoluenes and trimethylbenzenes) aromatics are important aromatic hydrocarbons in the atmosphere besides toluene and benzene (Monod et al., 2001; Millet et al., 2005; Heald et al., 2008; Kansal et al., 2009; Hu et al., 2015). The major sources of $C_8$ and $C_9$ aromatic hydrocarbons are fuel evaporation (Kaiser et al., 1992; Rubin et al., 2006; Miracolo et al., 2012), tailpipe exhaust (Singh et al., 1985; Monod et al., 2001; Lough et al., 2005; Na et al., 2005; Correa and Arbilla et al., 2006) and solvent use (Zhang et al., 2013). $C_8$ aromatic hydrocarbons (ethylbenzene and xylenes (ortho, meta and para) are categorized as hazardous air pollutants (HAPs) under the US Clean Air Act Amendments of 1990 (http://www.epa.gov/ttnatw01/orig189.html). Toluene and $C_8$ aromatics dominate anthropogenic SOA and SOA yield from all $C_9$ aromatics is currently predicted to be equal to that of toluene (Bahreini et al., 2009). The chemical composition of aromatic SOA remains poorly understood with less than 50 % of aromatic hydrocarbon photooxidation products identified (Forstner et al., 1997; Fisseha et al., 2004; Hamilton et al., 2005; Sato et al., 2007). Aromatic hydrocarbon photooxidation mechanisms remain uncertain except for the initial step ($\sim$ 90 % OH-addition reaction) (Calvert et al., 2002). Hence, understanding the atmospheric reaction mechanisms of $C_8$ and $C_9$ aromatic hydrocarbons and properly quantifying their SOA formation potential present unique challenges due to the variety in their molecular structure and the electron density of the aromatic ring.

Volatile organic compound (VOC) structure impacts the gas phase reaction mechanism (Ziemman and Atkinson, 2012) and kinetic reaction rate (e.g. $k_{OH}$ Atkinson, 1987) thereby influencing the resulting SOA properties and mass yield. Molecular structure impacts on SOA formation from alkanes have been previously studied (Lim and Ziemann, 2009; Ziemann, 2011; Lambe et al., 2012; Tkacik et al., 2012; Yee et al., 2013;

**ACPD**

doi:10.5194/acp-2015-871

**Impact of molecular structure on SOA formation**

L. Li et al.

**ACPD**

doi:10.5194/acp-2015-871

**Impact of molecular structure on SOA formation**

L. Li et al.

Loza et al., 2014). It is generally observed that SOA yield decreases from cyclic alkanes to linear alkanes and to branched alkanes. The relative location of the methyl group on the carbon chain also affects SOA yield (Tkacik et al., 2012). It is further found that the SOA yield and structure relationship is influenced by C=C groups (Ziemann, 2011). Understanding the SOA yield and structure relationship of aromatic compounds in a similar way is necessary due to the atmospheric importance of aromatic hydrocarbons.

Previously, aromatic studies categorized SOA yield solely based on substitute number (Odum, 1997a, b). However, those chamber experiments were conducted at high $NO_x$ conditions, which are well above levels present in the atmosphere. Song et al. (2005, 2007) found that initial $HC/NO_x$ ratios significantly impact SOA yields during aromatic photooxidation with yields increasing as $NO_x$ levels decreased. Ng et al. (2007) shows there is no significant yield difference between one substitute (toluene) and two substitute (*m*-xylene) aromatics in the absence of $NO_x$. The current work focuses on molecular structure impact on SOA formation at more atmospherically relevant $NO_x$ and aerosol loadings. Li et al. (2015a) demonstrated the methyl group number impact on SOA formation under low $NO_x$ conditions. Also, aromatic compounds with para position alkyl groups have been observed to form less SOA under various $NO_x$ conditions than their isomers in previous studies. Izumi and Fukuyama (1990) found that *p*-xylene, *p*-ethyltoluene and 1,2,4-trimethylbenzene have low SOA formation potential under high $NO_x$ conditions. Song et al. (2007) observed that *p*-xylene has the smallest SOA yield among all xylenes in the presence of $NO_x$. The relative methyl position to -OH in dimethyl phenols also impacts SOA yield in the absence of $NO_x$ (Nakao et al., 2011), while Song et al. (2007) observed no significant SOA yield difference between *o*-xylene and *p*-xylene under $NO_x$ free conditions. Moreover, previous studies mainly focused on the carbon number effect on SOA formation (Lim and Ziemann, 2009; Li et al., 2005a) and seldom addressed the substitute carbon length impact on VOC oxidation and hence SOA formation. Different percentages of similar compounds are found when the substitute carbon length on the aromatic ring changes

**ACPD**

doi:10.5194/acp-2015-871

**Impact of molecular structure on SOA formation**

L. Li et al.

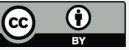

(Forstner et al., 1997; Huang et al., 2007, 2011). For example, a higher percentage of 3-methyl-2,5-furanone is observed in toluene than that of 3-ethyl-2,5-furanone in ethylbenzene (Forstner et al., 1997). Further, the branching structure on the aromatic substitute might impact the reaction pathway. It is possible that fragmentation is more favored on branched substitute alkoxy radicals than *n*-alkane substituents similar to alkanes (Atkinson et al., 2003).

Few studies comprehensively consider the overall alkyl effect on SOA formation from aromatic hydrocarbons, including the substitute number, position, carbon chain length and branching structure, especially under low $NO_x$ conditions. It is valuable to understand the relationship between aromatic hydrocarbon molecular structures and SOA physical and chemical characteristics. The effects of OH exposure (Lambe et al., 2011, 2015), mass loading (Shilling et al., 2009; Pfaffenberger et al., 2013) and NO condition (Ng et al., 2007; Eddingsaas et al., 2012) on SOA physical and chemical characteristics are previously discussed. However, few studies address the molecular structure effect of the precursor on SOA chemical composition, especially under atmospherically relevant conditions. Sato et al. (2012) shows the chemical composition difference between ethylbenzene, *m*-xylene, *o*-xylene, 1,2,4-trimethylbenzene and 1,3,5-trimethylbenzene under high absolute $NO_x$ conditions and hypothesizes that ketones prevent further oxidation during aromatic photooxidation compared with aldehydes. The SOA products detected in Sato's study are mainly small volatile compounds which are less likely to partition into the particle phase (Chhabra et al., 2011). Therefore, the study of Sato et al. (2012) indicates that further oxidation or oligomerization might contribute to SOA formation during aromatic photooxidation. Less SOA characterization data on propylbenzene and ethyltoluene compared with trimethylbenzene is available. However, Bahreini et al. (2009) suggests that the sum of the propylbenzene and ethyltoluene is on average a factor of 4–10 more abundant than trimethylbenzene.

This work examines twelve aromatic hydrocarbons, all of which are isomers with eight or nine carbons, to investigate the impact of molecular structure on SOA formation from aromatic hydrocarbon photooxidation under low $NO_x$. Here, we investigate the

substitute number, substitute position, alkyl carbon chain length and alkyl branching impacts on aromatic hydrocarbon oxidation.

## 2   Method

### 2.1   Environmental chamber

The UC Riverside/CE-CERT indoor dual $90\,m^3$ environmental chambers were used in this study and are described in detail elsewhere (Carter et al., 2005). Experiments were all conducted at dry conditions (RH < 0.1 %), in the absence of inorganic seed aerosol and with temperature controlled to $27 \pm 1\,°C$. Two movable top frames were slowly lowered during each experiment to maintain a slight positive differential pressure ($\sim 0.02''$ $H_2O$) between the reactors and enclosure to minimize dilution and/or contamination of the reactors. 272 115 W Sylvania 350BL blacklights are used as light sources for photooxidation.

A known volume of high purity liquid hydrocarbon precursors (ethylbenzene Sigma-Aldrich, 99.8 %; $n$-propylbenzene Sigma-Aldrich, 99.8 %; isopropylbenzene Sigma-Aldrich, analytical standard; $m$-xylene Sigma-Aldrich, 99 %; $o$-xylene Sigma-Aldrich, 99 %; $p$-xylene Sigma-Aldrich, 99 %; $m$-ethyltoluene Sigma-Aldrich, 99 %; $o$-ethyltoluene Sigma-Aldrich, 99 %; $p$-ethyltoluene Sigma-Aldrich, ≥ 95 %; 1,2,3-trimethylbenzene Sigma-Aldrich, OEKANAL analytical standard; 1,2,4-trimethylbenzene Sigma-Aldrich, 98 %; 1,3,5-trimethylbenzene Sigma-Aldrich, analytical standard) was injected through a heated glass injection manifold system and flushed into the chamber with pure $N_2$. NO was introduced by flushing pure $N_2$ through a calibrated glass bulb filled to a predetermined partial pressure of pure NO. All hydrocarbons and NO are injected and well mixed before lights are turned on to start the experiment.

Discussion Paper | Discussion Paper | Discussion Paper | Discussion Paper |

**ACPD**

doi:10.5194/acp-2015-871

**Impact of molecular structure on SOA formation**

L. Li et al.

## 2.2 Particle and gas measurement

Particle size distribution between 27 and 686 nm was monitored by dual custom built Scanning Mobility Particle Sizers (SMPS) (Cocker et al., 2001). Particle effective density was measured with an Aerosol Particle Mass Analyzer (APM-SMPS) system (Malloy et al., 2009). Particle volatility was measured by a Dekati® Thermodenuder Volatility Tandem Differential Mobility Analyzer (VTDMA) (Rader and McMurry, 1986) with a 17 s heating zone residence time (Qi et al., 2010a). The heating zone was controlled to 100 °C in this study with Volume fraction remaining (VFR) calculated as $(D_{p,\,after\,TD}/D_{p,\,before\,TD})^3$.

Particle-phase chemical composition evolution was measured by a High Resolution Time of Flight Aerosol Mass Spectrometer (HR-ToF-AMS; Aerodyne Research Inc.) (Canagaratna et al., 2007; DeCarlo et al., 2006). The sample was vaporized by a 600 °C oven followed by a 70 eV electron impact ionization. $f_x$ in this study is calculated as the fraction of the organic signal at $m/z = x$. For example, $f_{44}$, $f_{43}$, $f_{57}$ and $f_{71}$ are the ratios of the organic signal at $m/z$ 44, 43, 57 and 71 to the total organic signal, respectively (Chhabra et al., 2011; Duplissy et al., 2011). Elemental ratios for total organic mass, oxygen to carbon (O/C), and hydrogen to carbon (H/C) were determined using the elemental analysis (EA) technique (Aiken et al., 2007, 2008). Data was analyzed with ToF-AMS analysis toolkit squirrel 1.56D/PIKA 1.15D version.

The Agilent 6890 Gas Chromatograph – Flame Ionization Detector was used to measure aromatic hydrocarbon concentrations. A Thermal Environmental Instruments Model 42C chemiluminescence NO analyzer was used to monitor NO, $NO_y$-NO and $NO_y$. The gas-phase reaction model SAPRC-11 developed by Carter and Heo (2012) was utilized to predict radical concentrations (·OH, $HO_2$·, $RO_2$· and $NO_3$·).

Discussion Paper | Discussion Paper | Discussion Paper | Discussion Paper | Discussion Paper |

[ACPD](doi:10.5194/acp-2015-871)

doi:10.5194/acp-2015-871

**Impact of molecular structure on SOA formation**

L. Li et al.

**ACPD**

doi:10.5194/acp-2015-871

**Impact of molecular structure on SOA formation**

L. Li et al.

# 3 Result

## 3.1 SOA yield

Photooxidation of twelve $C_8$ and $C_9$ aromatic hydrocarbons were studied for low $NO_x$ conditions (HC/NO ratio 11.1–171 ppbC:ppb). SOA yields for all aromatic hydrocarbons were calculated according to Odum et al. (1996) as the mass ratio of aerosol formed to parent hydrocarbon reacted. Experimental conditions and SOA yields are listed (Table 1) along with additional *m*-xylene, *o*-xylene, *p*-xylene and 1,2,4-trimethylbenzene experimental conditions from previous studies (Song et al., 2005; Li et al., 2015a) (Table S2 in the Supplement). SOA yield as a function of particle mass concentration ($M_0$), shown in Fig. 1, includes experiments listed in both Tables 1 and S2. Each symbol represents a different aromatic hydrocarbon. It is observed that both alkyl substitute number and position affect SOA yield. The SOA yield of two-substitute $C_8$ and $C_9$ aromatic hydrocarbons depends more on the substitute location than substitute length. This means that the yield trend of *o*-xylene is analogous to that of *o*-ethyltoluene. Similarly, the yield trends for meta and para position substituted $C_8$ and $C_9$ aromatic hydrocarbons will be analogous to each other. Ortho isomers (*o*-xylene and *o*-ethyltoluene, marked as solid and hollow green circles, respectively) have the highest SOA yield for similar aerosol concentrations while para isomers (*p*-xylene and *p*-ethyltoluene, marked as solid and hallow blue diamonds, respectively) have the lowest SOA yield level. Lower SOA yield for para isomers are consistent with previous observation by Izumi and Fukuyama (1990). Izumi and Fukuyama (1990) also suggest that 1,2,4-trimethylbenzene yields are lower than for other aromatic hydrocarbons. The current study does not show a significant SOA yield difference between 1,2,4-trimethylbenzene and 1,3,5-trimethylbenzene. It is difficult to compare 1,2,3-trimethylbenzene yields with the former two trimethylbenzenes since 1,2,3-trimethylbenzene mass loading is much higher than the former two.

Aromatic hydrocarbons having only one substitute (ethylbenzene, *n*-propylbenzene and isopropylbenzene) or three substitutes (1,2,3-trimethylbenzene, 1,2,4-

Discussion Paper | Discussion Paper | Discussion Paper | Discussion Paper

trimethylbenzene and 1,3,5-trimethylbenzene) tend to have yields similar to the meta position two alkyl aromatics. Odum et al. (1997b) categorized SOA yield formation potential solely based on substitute number and claimed that aromatics with less than two methyl or ethyl substitutes form more particulate matter than those with two or more methyl or ethyl substitutes on the aromatic ring. However, Odum's work was conducted for high $NO_x$ conditions and had insufficient data to compare isomer yield differences (e.g., only two low mass loadings for $o$-xylene data). The strong low yield (two or more substitutes) and high yield (less than two methyl or ethyl substitutes) trends for high $NO_x$ conditions (Odum et al., 1997) are not observed for low $NO_x$ aromatic experiments in this study. Rather, high yield is observed only for benzene (Li et al., 2015a) while low yield is seen for substituted aromatic hydrocarbons. Similar SOA yield trends from different $C_8$ and $C_9$ aromatic isomers are further confirmed by comparing yields at similar radical conditions (Table S4, Fig. S2 in the Supplement). It is also found that molecular structure exerts a greater impact on SOA yield than reaction kinetics (Supplement). A two product model described by Odum et al. (1996) is used to fit SOA yield curves as a function of $M_0$. The twelve aromatics are categorized into five groups to demonstrate the alkyl group number and position effect on SOA formation. The five groups include one substitute group (1S), ortho position two alkyl group (ortho), meta position two alkyl group (meta), para position two alkyl group (para) and three substitute group (3S). Fitting parameters ($\alpha_1$, $K_{om,1}$, $\alpha_2$ and $K_{om,2}$; Table 2) in the two product model are determined by minimizing the sum of the squared residuals. The lower volatility partitioning parameter ($K_{om,2}$) is the same for all yield curve fits by assuming similar high volatile compounds are formed during all aromatic hydrocarbon photooxidation experiments. The ortho group is associated with a much higher $K_{om,1}$ compared with other aromatic groups, indicating aromatic hydrocarbon oxidation with an ortho position substitute forms much lower volatility products than other isomers. $K_{om,1}$ are also slightly higher in the meta group and one substitute groups than in the three substitute and para substitute groups.

**ACPD**

doi:10.5194/acp-2015-871

**Impact of molecular structure on SOA formation**

L. Li et al.

Discussion Paper | Discussion Paper | Discussion Paper | Discussion Paper

A slight SOA yield difference remains within each group (Fig. S1 and Table S3 in the Supplement), indicating the influence of factors other than alkyl group position. Generally, lower yields are found in aromatics with higher carbon number substitute alkyl groups, such as when comparing propylbenzene (*i*- and *n*-) with ethylbenzene or toluene (Li et al., 2005a), *m*-ethyltoluene with *m*-xylene and *p*-ethyltoluene with *p*-xylene, respectively. These differences are explained by the proposed alkyl group dilution effect (Sect. 4).

## 3.2  Chemical composition

### 3.2.1  $f_{44}$ vs. $f_{43+57+71}$

The ratio of alkyl substitute carbon number (H : C > 1) to the aromatic ring carbon number impacts SOA composition since the H : C ratio on the alkyl substitute is larger than 1 and the H : C ratio on aromatic ring itself is no more than 1. $m/z$ 43 ($C_2H_3O^+$ and $C_3H_7^+$) combined with $m/z$ 44 ($CO_2^+$) are critical to characterize oxygenated compounds in organic aerosol (Ng et al., 2010, 2011). $C_2H_3O^+$ is the major contributor to $m/z$ 43 in SOA formed from aromatic hydrocarbons having only methyl substitute (Li et al., 2015a) while $C_3H_7^+$ fragments are observed in this work for SOA from propylbenzene and isopropylbenzene (Fig. S4 in the Supplement). The $C_nH_{2n-1}O^+$ fragment in SOA corresponds to a $C_nH_{2n+1}$-alkyl substitute to the aromatic ring. $C_3H_5O^+$ ($m/z$ 57) and $C_4H_7O^+$ ($m/z$ 71) are important when investigating SOA from ethyl or propyl substitute aromatic precursors. While $m/z$ 43 ($C_3H_7^+$) and 57 ($C_4H_9^+$) are often considered as markers for hydrogen-like organic aerosol (Zhang et al., 2015; Ng et al., 2010), $C_3H_5O^+$ ($m/z$ 57) and $C_4H_7O^+$ ($m/z$ 71) are the major fragments (Fig. S4) from SOA originating from ethyl and propyl substituted aromatics, consistent with Sato et al. (2010) and Mohr et al. (2009). Figure S4 lists all fragments found at $m/z$ 43, 44, 57 and 71 and Fig. S5 in the Supplement shows the fraction of each $m/z$ in SOA formed from all aromatic hydrocarbons studied. The $m/z$ 43 + $m/z$ 44 + $m/z$ 57 + $m/z$ 71 accounts for 21.2 to 29.5 % of the total mass fragments from all $C_8$ and $C_9$ aromatics studied, suggesting

Discussion Paper | Discussion Paper | Discussion Paper | Discussion Paper

**ACPD**

doi:10.5194/acp-2015-871

**Impact of molecular structure on SOA formation**

L. Li et al.

similar oxidation pathways. Only a small fraction ($<\sim 0.7\,\%$) of $m/z$ 71 ($C_4H_7O^+$) or $m/z$ 57 ($C_3H_5O^+$) was observed in ethyltoluenes and trimethylbenzenes, respectively.

Figure S3 in the Supplement shows the evolution of $f_{44}$ and $f_{43+57+71}$ in SOA formed from the photooxidation of different aromatic hydrocarbons at low $NO_x$ conditions. $f_{44}$ and $f_{43+57+71}$ ranges are comparable to previous chamber studies (Ng et al., 2010; Chhabra et al., 2011; Loza et al., 2012; Sato et al., 2012). Only slight $f_{44}$ and $f_{43+57+71}$ evolution during chamber photooxidation is observed for the $C_8$ and $C_9$ isomers hence only the average $f_{44}$ and $f_{43+57+71}$ will be analyzed in this work.

A modification is applied to the mass based $m/z$ fraction in order to compare the mole relationship between $m/z$ 44 and $m/z$ 43 + $m/z$ 57 + $m/z$ 71 (Eq. 1).

$$f'_{43+57+71} = \frac{44}{43}f_{43} + \frac{44}{57}f_{57} + \frac{44}{71}f_{71} \qquad (1)$$

The average $f_{44}$ vs. $f'_{43+57+71}$ for all $C_8$ and $C_9$ isomers (Fig. 2) are located around the trend line for methyl group substituted aromatic hydrocarbons (Li et al., 2015a), implying a similarity in the SOA components formed from alkyl substituted aromatic hydrocarbons. A decreasing trend in oxidation from upper left to lower right is included in Fig. 2, similar to what Ng et al. (2011) found in the $f_{44}$ vs. $f_{43}$ graph, especially while comparing similar structure compounds. The methyl group location on the aromatic ring impacts $f_{44}$: $f'_{43+57+71}$. Decreasing $f_{44}$ and increasing $f'_{43+57+71}$ trends are observed from $p$-xylene to $o$-xylene to $m$-xylene and from 1,2,4-trimethylbenzene to 1,2,3-trimethylbenzene to 1,3,5-trimethylbenezene. The $f'_{43+57+71}$ may partially depend on the relative position between the alkyl substitute and the peroxide oxygen of the bicyclic hydroperoxide. For instance, allylically stabilized five-membered bicyclic radicals are the most stable bicyclic peroxide formed from aromatic hydrocarbon photooxidation (Andino et al., 1996). Two meta position substitutes connected to the aromatic ring carbon with -C-O- yield higher fractions of $C_nH_{2n-1}O^+$ fragments than the para and ortho position, which have at most one substitute connected with -C-O-. More importantly, the difference in $f_{44}$ implies that substitute location influences the further reaction

**ACPD**

doi:10.5194/acp-2015-871

**Impact of molecular structure on SOA formation**

L. Li et al.

pathway to form $CO_2^+$ since $CO_2^+$ is not readily available from bicyclic hydroperoxides. This indicates that the alkyl groups are more likely to contribute to SOA formation at the meta position than the ortho and para position. The para position substituted aromatics form the least SOA as they exclude the possibility that para position alkyl substitutes are further oxidized to other less volatile components instead of $C_nH_{2n-1}O^+$. *p*-Xylene displays the high $f_{44}$ similar to benzene (Li et al., 2015a) implying that the para position substitute exerts the least dilution effect on $CO_2^+$ formation pathway among all isomers. Bicyclic hydroperoxides formed from the OH-addition reaction pathway and their dissociation reaction products are both used to explain the substitute location impact on $f_{44}$ and $f'_{43+57+71}$ relationship. However, the existence of longer alkyl substitutes diminishes the alkyl substitute location impact. SOA $f_{44}$ and $f'_{43+57+71}$ in ethyltoluenes are all analogous to *m*-xylene. One substitute $C_8$ and $C_9$ aromatic hydrocarbons have similar $f_{44}$ and $f'_{43+57+71}$ with slightly lower $f_{44}$ and $f'_{43+57+71}$ compared to toluene (Li et al., 2015a). Longer alkyl substitutes may not lower the average oxidation per mass as further oxidation of the longer chain alkyls may render other oxidized components not included in Fig. 2. Their lower total $f_{44} + f'_{43+57+71}$ (Fig. S5) further supports the possibility of oxidation of the longer alkyl substitutes. Elemental ratio (Sect. 3.2.2) and oxidation state (Sect. 3.2.3) are further used to evaluate the impact of increasing alkyl group size on SOA formation.

## 3.2.2  H/C vs. O/C

Elemental analysis (Aiken et al., 2007, 2008) serves as a valuable tool to elucidate SOA chemical composition and SOA formation mechanisms (Heald et al., 2010; Chhabra et al., 2011). Figure S6 in the Supplement shows H/C and O/C evolution in SOA formed from the photooxidation of different aromatic hydrocarbons under low NO$_x$ (marked and colored similarly to Fig. S3). H/C and O/C ranges are comparable to previous chamber studies (Chhabra et al., 2011 (*m*-xylene and toluene); Loza et al., 2012 (*m*-xylene); Sato et al., 2012 (benzene and 1,3,5-trimethylbenzene)). SOA components from all isomers are located in between slope = −1 and slope = −2 lines (Fig. S6) sug-

**[ACPD](doi:10.5194/acp-2015-871)**

doi:10.5194/acp-2015-871

**Impact of molecular structure on SOA formation**

L. Li et al.

Discussion Paper | Discussion Paper | Discussion Paper | Discussion Paper

gesting that SOA from these aromatic hydrocarbons is composed primarily of acid (carbonyl acid and hydroxycarbonyl) and carbonyl (ketone or aldehyde) like functional groups. The elemental ratio of SOA from $p$-xylene photooxidation was nearly located on the acid line (slope = −1). The SOA elemental ratio for $C_8$ and $C_9$ aromatic iso-

mers are located near the alkyl number trend line found by Li et al. (2015a) for methyl substituents, indicating a similarity between SOA from various alkyl substituted hydrocarbons. SOA formed is among the LV-OOA and SV-OOA regions (Ng et al., 2011). The evolution trend agrees with Fig. S3 (Sect. 3.2.1). The current study concentrated on experimentally averaged H/C and O/C to explore the impact of molecular structure

on SOA chemical composition.

Average H/C and O/C locations are marked with aromatic compound names in Fig. 3. All H/C and O/C are located around the predicted values for $C_8$ and $C_9$ SOA (dark solid circle) based on the elemental ratio of benzene SOA (Li et al., 2015a). This confirms the presence of a carbon dilution effect in all isomers. Ortho position aro-

matic hydrocarbons (*o*-xylene or *o*-ethyltoluene) lead to a more oxidized SOA (higher O/C and lower H/C) than that of meta (*m*-xylene or *m*-ethyltoluene) and para (*p*-xylene or *p*-ethyltoluene) aromatics. SOA formed from 1,2,4-trimethylbenzene and 1,2,3-trimethylbenzene is more oxidized than that from 1,3,5-trimethylbenzene. It is noticed that 1,2,4-trimethylbenzene and 1,2,3-trimethylbenzene both contain an ortho

position moiety on the aromatic ring. This indicates that the ortho position aromatic hydrocarbon is readily oxidized and this ortho position impact on oxidation extends to triple substituted aromatic hydrocarbons.

Substitute length also plays an important role in aromatic hydrocarbon oxidation. Overall, SOA from a one-substitute aromatic with more carbon in the substitute is lo-

cated at a more oxidized area of the O/C vs. H/C chart (lower right in Fig. 3.) than those multiple substitute aromatic isomers with the same total number of carbon as the single substituted aromatic. SOA from isopropylbenzene is located in a lower position of the chart and to the right of propylbenzene indicating that branch carbon structure on the alkyl substitute of aromatic hydrocarbons leads to a more oxidized SOA. Lines

**[ACPD](doi:10.5194/acp-2015-871)**

doi:10.5194/acp-2015-871

**Impact of molecular structure on SOA formation**

L. Li et al.

in Fig. S7 in the Supplement connect the O/C and H/C of resulting SOA to that of the aromatic precursor. Most SOA components show a slight H/C increase and a dramatic O/C increase from the precursor, which is consistent with results observed for methyl substituted aromatics (Li et al., 2015a). However, H/C barely increases (1.33 to 1.34) from the propylbenzene precursor to its resulting SOA and there is even a decreasing trend from isopropylbenzene to its SOA. This indicates that a high H/C component loss reaction such as alkyl part dissociation during photooxidation is an important reaction to SOA formation from longer carbon chain containing aromatic hydrocarbons. The carbon chain length of propylbenzene increases the possibility of alkyl fragmentation. The branching structure of isopropylbenzene facilitates fragmentation through the stability of tertiary alkyl radicals. Elemental ratio differences between xylenes and ethyltoluenes can be attributed to the alkyl dilution effect, similar to the methyl dilution theory by Li et al. (2015a). *m*-Ethyltoluene is an exception as it is more oxidized than *m*-xylene after accounting for the alkyl dilution effect. This suggests that the meta substituted longer chain alkyl is more readily oxidized since the substitute is at the end of ring opening products. Prediction of elemental ratios from toluene and xylenes are discussed later (Sect. 4) to further quantify the carbon length and branching effect on SOA formation from aromatic hydrocarbons.

### 3.2.3 OS$_c$

Oxidation state (OS$_c$ ≈ 2O/C-H/C) was introduced into aerosol phase component analysis by Kroll et al. (2011). It is considered to be a more accurate metric for describing oxidation in atmospheric organic aerosol (Ng et al., 2009; Canagaratna et al., 2015; Lambe et al., 2015) and therefore well correlated with gas-particle partitioning (Aumont et al., 2012;). Average OS$_c$ of SOA formed from C$_8$ and C$_9$ aromatic isomers ranges from −0.53 to −0.20 and −0.82 to −0.22, respectively (Fig. 4), implying that the precursor molecular structure impacts the OS$_c$ of the resulting SOA. An OS$_c$ decrease with alkyl substitute length is observed in one-substitute aromatic hydrocarbons from toluene (toluene OS$_c$ = −0.049; Li et al., 2015a) to propylben-

**ACPD**

doi:10.5194/acp-2015-871

**Impact of molecular structure on SOA formation**

L. Li et al.

**ACPD**

doi:10.5194/acp-2015-871

**Impact of molecular structure on SOA formation**

L. Li et al.

zene. However, $OS_c$ provides the average oxidation value per carbon not considering whether these carbons start from an aromatic ring carbon or an alkyl carbon. Alkyl carbons are associated with more hydrogen than aromatic ring carbons, thus leading to a lower precursor $OS_c$ and therefore lower SOA $OS_c$. Dilution theory in Sect. 4 will be used to further explore the carbon chain length effect on aromatic hydrocarbon oxidation by considering the precursor H : C ratio. Single substitute aromatic hydrocarbons generally show higher $OS_c$ than multiple substitute ones, consistent with the yield trend of Odum et al. (1997b). However, it is also found that ortho position moiety containing two or three substitute aromatic hydrocarbons have analogous $OS_c$ to single substitute aromatic hydrocarbons (*o*-xylene −0.202 to ethylbenzene −0.197; 1,2,4-trimethylbenzene −0.433 and *o*-ethyltoluene −0.492 to propylbenzene −0.435). This suggests that both substitute number and position are critical to aromatic hydrocarbon oxidation and therefore SOA formation. $OS_c$ trends also support that the meta position suppresses oxidation while the ortho position promotes oxidation when the $OS_c$ of xylenes (*o*-xylene > *p*-xylene > *m*-xylene), ethyltoluenes (*o*-ethyltoluene > *p*-ethyltoluene > *m*-ethyltoluene) and trimethylbenzenes (1,2,4-trimethylbenzene (ortho moiety containing) > 1,2,3-trimethylbenzene (ortho moiety containing) > 1,3,5-trimethylbenzene (meta moiety containing)) are compared separately. Further, SOA formed from isopropylbenzene shows the highest $OS_c$ among all $C_9$ isomers, nearly equivalent to that of ethylbenzene. This demonstrates that the branching structure of the alkyl substitute can enhance further oxidation of aromatic hydrocarbons.

## 3.3  Physical property

### 3.3.1  SOA density

SOA density is a fundamental parameter in understanding aerosol morphology, dynamics, phase and oxidation (De Carol et al., 2004; Katrib et al., 2005; Dinar et al., 2006; Cross et al., 2007). SOA density ranges from 1.29–1.38 g cm$^{-3}$ from aromatic photooxidation under low NO$_x$ conditions in this study (Fig. 5). The range is comparable to

Discussion Paper | Discussion Paper | Discussion Paper | Discussion Paper |

previous studies under similar conditions (Esther Borrás and Tortajada-Genaro, 2012; Ng et al., 2007; Sato et al., 2010). There is no significant difference in the density of SOA formed from $C_8$ and $C_9$ aromatic hydrocarbon isomers and molecular structure is not observed to be a critical parameter to determine SOA density. The standard deviation results from differences in initial conditions (e.g., initial HC/NO) that also determine the oxidation of aromatic hydrocarbons (Li et al., 2015b) and thus further affect density. SOA density is best correlated with the O/C ratio and $OS_c$ (0.551 and 0.540, Table 3), consistent with the observation of Pang et al. (2006) that SOA density increases with increasing O/C ratio. The density prediction method developed by Kuwata et al. (2011) based on O/C and H/C is evaluated as

$$\rho = \frac{12 + \text{H/C} + 16 \times \text{O/C}}{7 + 5 \times \text{H/C} + 4.15 \times \text{O/C}} \tag{2}$$

The black lines (Fig. 5) are predicted (Eq. 2) densities and show a good agreement between predicted and measured SOA densities (−6.01 to 6.90 %).

### 3.3.2   SOA volatility

SOA volatility is associated with reactions such as oxidation, fragmentation, oligomerization and mass loading (Kalberer et al., 2004; Salo et al., 2011; Tritscher et al., 2011; Yu et al., 2014). SOA volatility in this study is measured as VFR. Initial (< 30 min after new particle formation) SOA VFRs are around 0.2 for all the aromatic precursors studied and increase up to 0.58 during photooxidation. This suggests that aromatic hydrocarbon oxidation undergoes an evolution from volatile compounds to semivolatile compounds. The VFR trends and ranges are comparable to previous studies (Kalberer et al., 2004; Qi et al., 2010a, b; Nakao et al., 2012). Figure 6 shows the VFR at the end of aromatic hydrocarbon photooxidation (VFR$_{end}$). A decreasing VFR$_{end}$ trend is found as the number of substitutes increase and for meta position (e.g. *m*-xylene) or meta position containing (e.g. 1,3,5-trimethylbenzene) aromatic precursors. Strong correlations

Discussion Paper | Discussion Paper | Discussion Paper | Discussion Paper |

**ACPD**

doi:10.5194/acp-2015-871

**Impact of molecular structure on SOA formation**

L. Li et al.

among VFR$_{end}$ and chemical composition are observed in the aromatic hydrocarbons studied here (Table 3). This is consistent with recent findings that O : C ratio is well correlated to aerosol volatility (Sect. 3.3.2) (Cappa et al., 2012; Yu et al., 2014), thereby affecting the gas-particle partitioning, which in turn relates to SOA yield. It is also observed that VFR$_{end}$ is strongly correlated (−0.937) with reaction rate constant ($k_{OH}$). Higher $k_{OH}$ is associated with faster reaction rates of initial aromatic precursors and is therefore expected to lead to further oxidation for a given reaction time. However, the inverse correlation between $k_{OH}$ and VFR$_{end}$ indicates that $k_{OH}$ value represents more than just the kinetic aspects. $k_{OH}$ increases with increasing number of substitutes on the aromatic ring. Additionally, aromatic hydrocarbons with meta position substitutes have higher $k_{OH}$ than those with para and ortha (Table S1 in the Supplement) position substitutes. This suggests that the precursor molecular structures for aromatics associated with $k_{OH}$ values determine the extent of oxidation of the hydrocarbons and therefore impact SOA volatility more than simply the precursor oxidation rate.

## 4   Alkyl dilution theory on SOA formation from aromatic hydrocarbons

The dependence of SOA formation on molecular structure can be partially represented by the alkyl carbon number. Methyl dilution theory (Li et al., 2015a) is extended to alkyl substitute dilution theory in order to investigate the influence of longer alkyl substitutes compared with methyl group substitutes. Figure 7a shows the predicted elemental ratio for SOA formed from longer alkyl substitutes (-C$_2$H$_{2n+1}$, $n > 1$) based on methyl only substitute. The elemental ratio of SOA formed from single substitute aromatic hydrocarbons including ethylbenzene, propylbenzene and isopropylbenzene are predicted by toluene and those of ethyltoluenes are predicted by corresponding xylenes with similar alkyl substitute location. H/C and O/C are generally overestimated by only considering alkyl dilution effect. However, OS$_c$ prediction is close to measurement (> ±15 %; Fig. 7b and dashed line in Fig. 7a) for ethyltoluenes. This suggests that higher carbon number alkyl substitutes may suppress reactions that have little effect on OS$_c$ but large effect

Discussion Paper | Discussion Paper | Discussion Paper | Discussion Paper |

**ACPD**

doi:10.5194/acp-2015-871

**Impact of molecular structure on SOA formation**

L. Li et al.

on elemental ratio (e.g hydrolysis). $OS_c$ is largely underestimated in SOA formed from single substitute aromatic hydrocarbons, especially for isopropylbenzene and ethylbenzene. This implies that longer alkyl substitutes are more oxidized than the methyl group on toluene. A direct ·OH reaction with the alkyl part of the aromatic is more favored on longer alkyl chains since tertiary and secondary alkyl radicals are more stable than primary alkyl radicals (Forstner et al., 1997). The less significant $OS_c$ underestimation from xylenes to ethyltoluenes is due to the presence of an "inert" methyl group which lowers the average $OS_c$. The extreme low H/C in isopropylbenzene implies an additional hydrogen loss with the branching alkyl substitute, which possibly occurs while forming 2,5-furandione (Forstner et al., 1997) or 3-H-furan-2-one due to the increased stability of the isopropyl radical compared to the *n*-propyl radical. It is also possible that longer carbon chain substitutes might have higher probability to form other cyclic or low vapor pressure products by additional reaction due to their increased length. The similarity in $f_{44}$ and $f'_{43+57+71}$ but discrepancy in elemental ratio among all single substitute $C_8$ and $C_9$ aromatics supports that additional reactions leading to further oxidization of alkyl substitutes can occur.

## 5  Atmospheric implication

This study elucidates molecular structure impact on a major anthropogenic SOA source, photooxidation of aromatic hydrocarbons, under atmospherically relevant $NO_x$ conditions by analyzing SOA yield, chemical composition and physical properties. These observations, when taken together, indicate the roles of alkyl substitute number, location, carbon chain length and branching structure in aromatic hydrocarbon photooxidation. SOA yield of all $C_8$ and $C_9$ aromatic hydrocarbon isomers are comprehensively provided in this study with a focus on the impact of molecular structure. It is demonstrated that aromatic hydrocarbon oxidation and SOA formation should not be simply explained by substitute number. The promoting of SOA formation by the ortho position is found along with confirmation of the suppression effect by the para

Discussion Paper | Discussion Paper | Discussion Paper | Discussion Paper |

**ACPD**

doi:10.5194/acp-2015-871

**Impact of molecular structure on SOA formation**

L. Li et al.

position during oxidation of aromatic hydrocarbons. Meta position alkyl substitutes on aromatic ring lead to a lower extend of aromatic hydrocarbon oxidation. Aromatic oxidation is proved to increase with alkyl substitute chain length and branching structure. Further, carbon dilution theory developed by Li et al. (2015a) is extended to this study. Carbon dilution theory not only serves as a tool to explain the difference in SOA components due to the difference in substitute alkyl carbon number but also acts as a standard to determine the oxidation mechanism based on alkyl substitute structure. This study improves the understanding of SOA formation from aromatic hydrocarbons and contributes to more accurate SOA prediction from aromatic precursors. Further study is warranted to reveal the detailed oxidation pathway of aromatic hydrocarbons with longer (carbon number > 1) alkyl substitutes.

*Acknowledgements.* We acknowledge funding support from National Science Foundation (ATM 0901282) and W. M. Keck Foundation. Any opinions, findings, and conclusions expressed in this material are those of the author(s) and do not necessarily reflect the views of the NSF.

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

**Table 1.** Experiment conditions.

| Precursor | ID | HC/NO[a] | NO[b] | HC[b] | ΔHC[c] | Mo[c] | Yield |
|---|---|---|---|---|---|---|---|
| Ethylbenzene | 1142A | 17.0 | 47.4 | 101 | 331 | 22.0 | 0.066 |
| | 1142B | 12.0 | 66.6 | 99.9 | 341 | 4.40 | 0.013 |
| | 1146A | 35.6 | 22.2 | 99.0 | 257 | 36.0 | 0.140 |
| | 1146B | 23.0 | 34.8 | 100 | 331 | 23.6 | 0.071 |
| | 1147B | 74.9 | 36.5 | 342 | 626 | 88.1 | 0.141 |
| | 2084A | 81.1 | 23.9 | 242 | 374 | 54.0 | 0.145 |
| | 2084B | 93.8 | 20.3 | 238 | 266 | 44.3 | 0.167 |
| Propylbenzene | 1245A | 41.0 | 22.1 | 101 | 231 | 11.8 | 0.051 |
| | 1246A | 26.8 | 68.5 | 204 | 421 | 22.9 | 0.054 |
| Isopropylbenzene | 1247A | 40.3 | 22.4 | 100 | 301 | 33.2 | 0.110 |
| | 1247B | 18.6 | 48.1 | 99.3 | 300 | 16.6 | 0.055 |
| | 1253A | 31.9 | 56.4 | 200. | 538 | 53.1 | 0.099 |
| | 1253B[d] | 17.6 | 100 | 196 | 526 | 16.5 | 0.031 |
| $o$-Xylene | 1315A | 13.2 | 49.8 | 82.2 | 324 | 26.3 | 0.081 |
| | 1315B | 28.8 | 22.2 | 80.0 | 27 | 25.4 | 0.091 |
| | 1320A | 12.8 | 50.0 | 80.0 | 335 | 18.4 | 0.055 |
| | 1321A | 31.0 | 20.5 | 79.2 | 263 | 16.2 | 0.061 |
| | 1321B | 61.3 | 10.4 | 80.0 | 226 | 9.80 | 0.044 |
| $p$-Xylene | 1308A | 15.5 | 55.6 | 78.4 | 279 | 6.80 | 0.024 |
| | 1308B | 171 | 22.9 | 78.8 | 274 | 11.3 | 0.041 |
| $m$-Ethyltoluene | 1151A | 17.9 | 62.5 | 84.8 | 409 | 8.30 | 0.020 |
| | 1151B | 31.0 | 32.3 | 86.4 | 415 | 28.7 | 0.069 |
| | 1199A | 8.8 | 45.4 | 100.2 | 447 | 72.0 | 0.161 |
| | 1222B | 41.7 | 69.4 | 100.0 | 484 | 70.9 | 0.146 |
| | 1226B | 11.3 | 137.6 | 201.1 | 895 | 138 | 0.154 |
| | 1232A | 27.5 | 122.0 | 200.0 | 901 | 150 | 0.167 |
| | 1232B | 33.1 | 67.5 | 194.8 | 751 | 117 | 0.155 |
| | 1421A | 41.0 | 22.1 | 97.9 | 409 | 46.2 | 0.112 |
| | 1421B | 18.0 | 44.9 | 98.7 | 477 | 54.6 | 0.114 |

**ACPD**

doi:10.5194/acp-2015-871

**Impact of molecular structure on SOA formation**

L. Li et al.

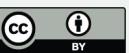

| Precursor | ID | HC/NO[a] | NO[b] | HC[b] | ΔHC[c] | Mo[c] | Yield |
|---|---|---|---|---|---|---|---|
| *o*-Ethyltoluene | 1179A | 16.3 | 52.9 | 91.7 | 399 | 86.5 | 0.216 |
| | 1179B | 15.8 | 52.9 | 93.0 | 415 | 75.3 | 0.181 |
| | 1202A | 18.5 | 60.3 | 99.7 | 422 | 69.9 | 0.166 |
| | 1215A | 29.2 | 107.9 | 180.3 | 637 | 1501 | 0.237 |
| | 1413A | 12.2 | 21.3 | 100.4 | 371 | 64.5 | 0.174 |
| | 1413B | 24.1 | 45.8 | 98.4 | 455 | 64.4 | 0.141 |
| *p*-Ethyltoluene | 1194A | 19.9 | 90.7 | 196 | 741 | 90.4 | 0.122 |
| | 1194B | 13.0 | 88.4 | 200 | 761 | 73.0 | 0.096 |
| | 1197A | 13.1 | 56.4 | 192 | 653 | 66.4 | 0.102 |
| | 1197B | 14.8 | 98.5 | 192 | 710 | 58.4 | 0.082 |
| | 1214B | 26.0 | 53.4 | 102 | 418 | 29.1 | 0.069 |
| | 1601A | 39.9 | 31.2 | 109 | 452 | 17.6 | 0.039 |
| 1,2,3-Trimethylbenzene | 1158A | 19.8 | 10.3 | 79.9 | 296 | 22.2 | 0.075 |
| | 1158B | 15.6 | 22.4 | 79.9 | 379 | 32.3 | 0.085 |
| | 1162A | 15.8 | 33.4 | 80.1 | 391 | 46.5 | 0.119 |
| | 1162B | 14.9 | 40.0 | 80.4 | 399 | 46.6 | 0.117 |
| 1,3,5-Trimethylbenzene | 1153A | 65.2 | 11.0 | 79.5 | 309 | 12.4 | 0.040 |
| | 1153B | 35.3 | 20.4 | 80 | 381 | 19.6 | 0.051 |
| | 1156A | 22.3 | 32.3 | 80.2 | 379 | 24.8 | 0.065 |
| | 1156B | 15.5 | 46.1 | 79.6 | 390 | 19.0 | 0.049 |
| | 1329B | 11.1 | 64.8 | 80.0 | 296 | 3.00 | 0.007 |

Note: [a] unit of HC/NO are ppbC : ppb; [b] unit of NO and HC are ppb; [c] unit of ΔHC and $M_0$ are $\mu g\,m^{-3}$, $M_0$ is a wall loss and density corrected particle mass concentration; [d] not used in curve fitting.

**ACPD**

doi:10.5194/acp-2015-871

**Impact of molecular structure on SOA formation**

L. Li et al.

**Table 2.** Two product yield curve fitting parameters for one, two (ortha, meta and para) and three alkyl substitutes.

| Yield Curve | $\alpha_1$ | $K_{om,1}$ $(m^3\,\mu g^{-1})$ | $\alpha_2$ | $K_{om,2}$ $(m^3\,\mu g^{-1})$ |
|---|---|---|---|---|
| One Substitutes | 0.144 | 0.039 | 0.137 | 0.005 |
| Two Substitutes-ortho | 0.158 | 0.249 | 0.024 | 0.005 |
| Two Substitutes-meta | 0.156 | 0.040 | 0.080 | 0.005 |
| Two Substitutes-para | 0.154 | 0.025 | 0.036 | 0.005 |
| Three Substitutes | 0.180 | 0.025 | 0.052 | 0.005 |

Discussion Paper | Discussion Paper | Discussion Paper | Discussion Paper

Discussion Paper | Discussion Paper | Discussion Paper | Discussion Paper | Discussion Paper

**ACPD**

doi:10.5194/acp-2015-871

**Impact of molecular structure on SOA formation**

L. Li et al.



**Table 3.** Correlation among SOA density, volatility (VFR) and SOA chemical composition.

| | $f_{44}$ | $f_{57}$ | $f_{71}$ | O/C | H/C | $OS_c$ | $k_{OH}$ |
|---|---|---|---|---|---|---|---|
| Density | 0.324 | −0.056 | −0.38 | 0.551 | −0.301 | 0.540 | −0.249 |
| $p$ value[b] | 0.304 | 0.862 | 0.223 | 0.063 | 0.341 | 0.070 | 0.435 |
| $VFR_{end}^a$ | 0.537 | 0.56 | 0.399 | 0.471 | −0.586 | 0.593 | −0.937 |
| $p$ value[b] | 0.089 | 0.073 | 0.224 | 0.144 | 0.058 | 0.055 | 0 |

Note: [a] $VFR_{end}$ volume fraction remaining at the end of photooxidation; [b] $p$ values range from 0 to 1, 0-reject null hypothesis and 1 accept null hypothesis. Alpha ($\alpha$) level used is 0.05. If the $p$ value of a test statistic is less than alpha, the null hypothesis is rejected.

**ACPD**

doi:10.5194/acp-2015-871

**Impact of molecular structure on SOA formation**

L. Li et al.

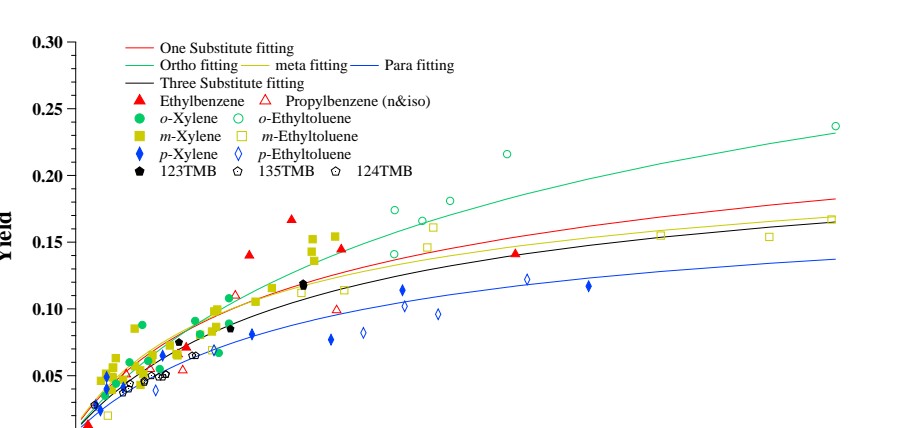

**Figure 1.** Aromatic SOA yields as a function of $M_0$. (Note: Song et al., 2005, 2007; Li et al., 2015a data are also included; 123TMB – 1,2,3-Trimethylbenzene; 135TMB – 1,3,5-Trimethylbenzene; 124TMB – 1,2,4-Trimethylbenzene.)

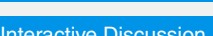


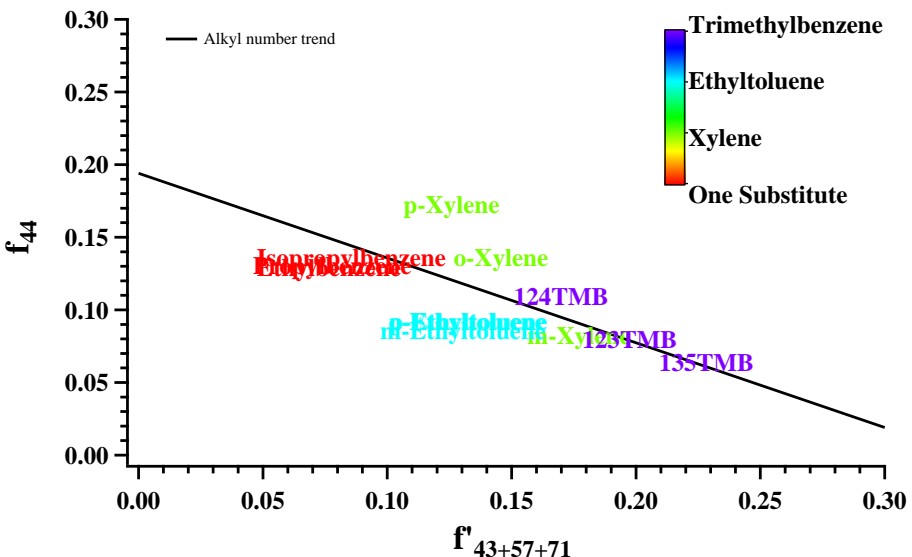

**Figure 2.** $f_{44+}$ vs. $f'_{43+57+71}$ in SOA formed from different aromatic hydrocarbon photoox-idation under low NO$_x$ colored by aromatic isomer type and marked with individual aro-matic hydrocarbon species: Ethylbenzene 2084A; Propylbenzene 1245A; Isopropylbenzene 1247A; $m$-Xylene 1191A; $m$-Ethyltoluene 1199A; $o$-Xylene 1320A; $o$-Ethyltoluene 1179A; $p$-Xylene 1308A; $p$-Ethyltoluene 1194A; 1,2,3-Trimethylbenzene (123TMB) 1162A; 1,2,4-Trimethylbenzene (124TMB) 1119A; 1,3,5-Trimethylbenzene (135TMB) 1156A. Alkyl number trend is the linear fitting in (Li et al., 2015a).

Discussion Paper | Discussion Paper | Discussion Paper | Discussion Paper

# ACPD

doi:10.5194/acp-2015-871

**Impact of molecular structure on SOA formation**

L. Li et al.

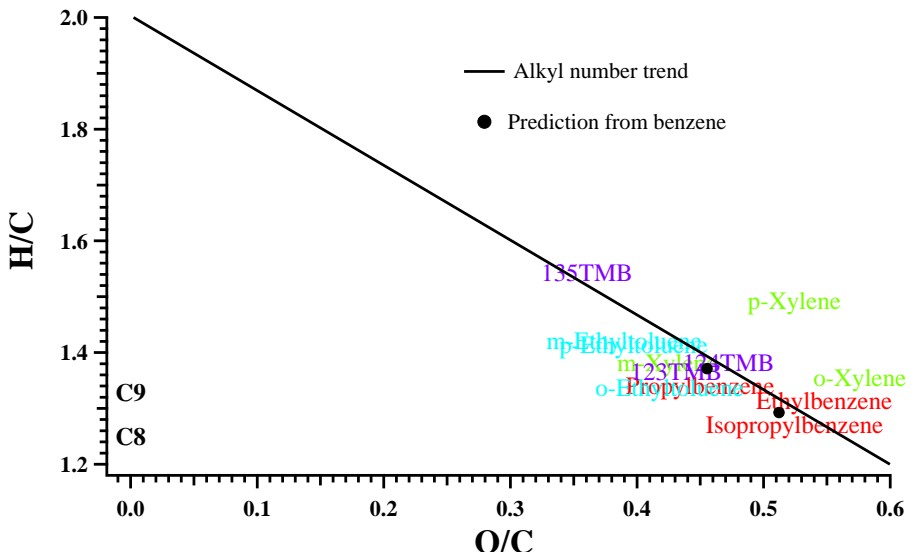

**Figure 3.** H/C vs. O/C in SOA formed from different aromatic hydrocarbon photooxi-dation under low NO$_x$ colored by aromatic isomer type and marked with individual aro-matic hydrocarbon species (C8 and C9 on the lower left indicate the location of initial aromatic hydrocarbon precursor): Ethylbenzene 2084A; Propylbenzene 1245A; Isopropyl-benzene 1247A; $m$-Xylene 1191A; $m$-Ethyltoluene 1199A; $o$-Xylene 1320A; $o$-Ethyltoluene 1179A; $p$-Xylene 1308A; $p$-Ethyltoluene 1194A; 1,2,3-Trimethylbenzene (123TMB) 1162A; 1,2,4-Trimethylbenzene(124TMB) 1119A; 1,3,5-Trimethylbenzene(135TMB) 1156A. Alkyl num-ber trend is the linear fitting in (Li et al., 2015a). Solid black cycles are SOA elemental ratio from C$_8$ and C$_9$ aromatic hydrocarbon predicted by SOA elemental ratio formed from benzene.

Discussion Paper | Discussion Paper | Discussion Paper | Discussion Paper |

**ACPD**

doi:10.5194/acp-2015-871

**Impact of molecular structure on SOA formation**

L. Li et al.

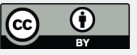

Discussion Paper | Discussion Paper | Discussion Paper | Discussion Paper

# ACPD

doi:10.5194/acp-2015-871

**Impact of molecular structure on SOA formation**

L. Li et al.



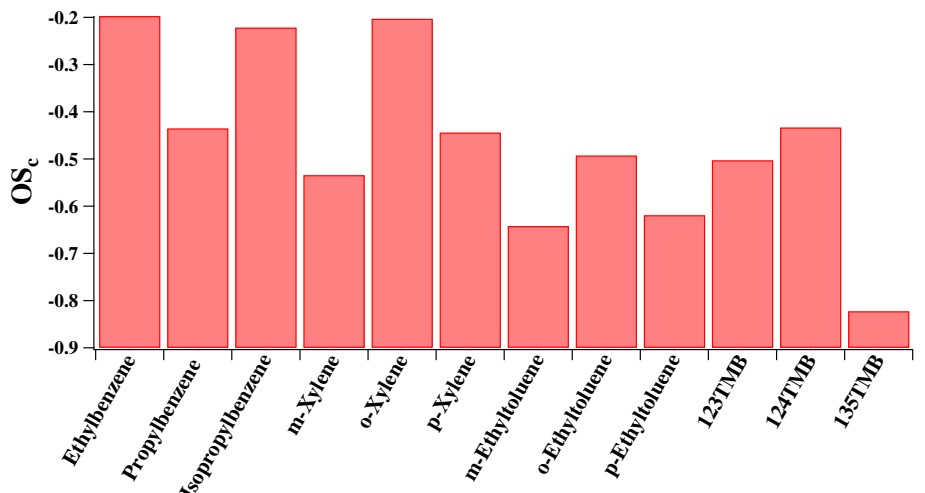

**Figure 4.** Oxidation state ($OS_c$) of SOA formed from different aromatic hydrocarbon photooxidation under low $NO_x$: Ethylbenzene 2084A; Propylbenzene 1245A; Isopropylbenzene 1247A; *m*-Xylene 1191A; *m*-Ethyltoluene 1199A; *o*-Xylene 1320A; *o*-Ethyltoluene 1179A; *p*-Xylene 1308A; *p*-Ethyltoluene 1194A; 1,2,3-Trimethylbenzene (123TMB) 1162A; 1,2,4-Trimethylbenzene(124TMB) 1119A; 1,3,5-Trimethylbenzene(135TMB) 1156A.

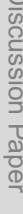

Discussion Paper | Discussion Paper | Discussion Paper | Discussion Paper

# ACPD

doi:10.5194/acp-2015-871

**Impact of molecular structure on SOA formation**

L. Li et al.

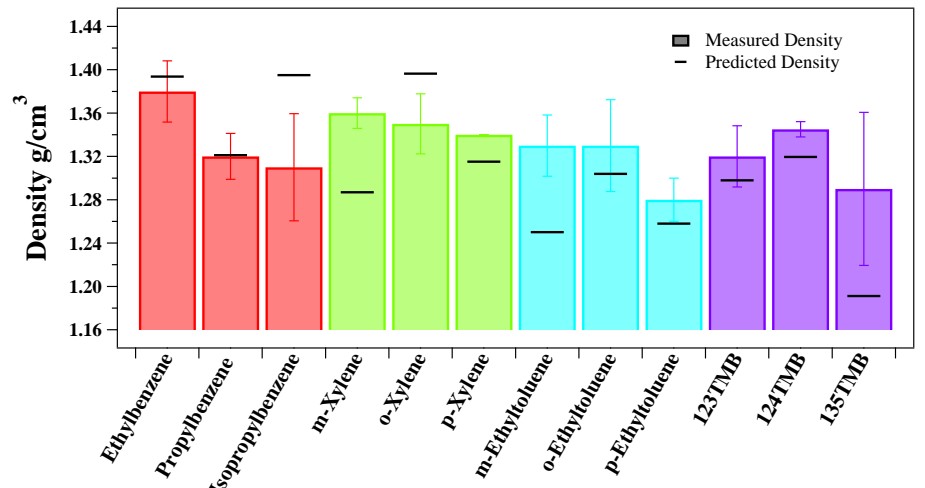

**Figure 5.** Measured and predicted SOA density from different aromatic hydrocarbon photooxidation under low NO$_x$ (Colored with substitute number and length, one substitute-red, xylenes-green, ethyltoluenes-blue and trimethylbenzene-purple; black line is predicted density according to Kuwata et al., 2011); 123TMB – 1,2,3-Trimethylbenzene; 135TMB – 1,3,5-Trimethylbenzene; 124TMB – 1,2,4-Trimethylbenzene.

**ACPD**

doi:10.5194/acp-2015-871

**Impact of molecular structure on SOA formation**

L. Li et al.

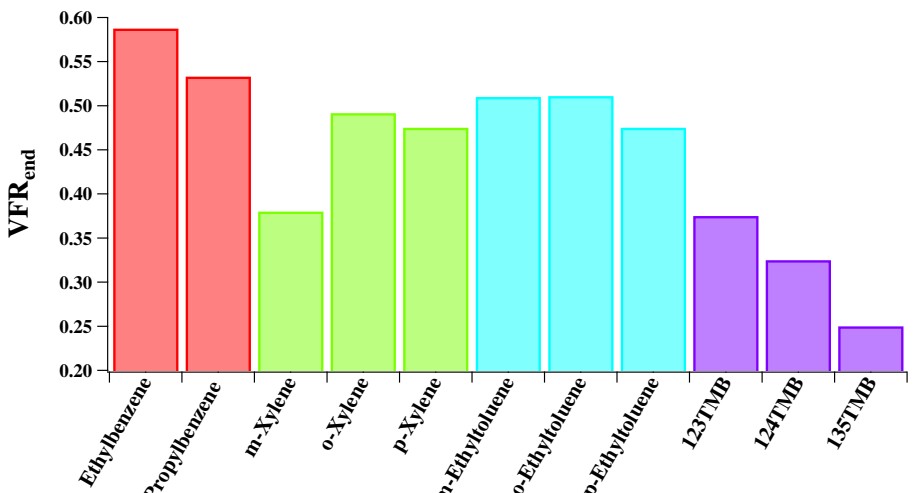



**Figure 6.** SOA Volume fraction remaining ($VFR_{end}$) at the end of aromatic hydrocarbon photooxidation under low $NO_x$ (Colored with substitute number and length, one substitute-red, xylenes-green, ethyltoluenes-blue and trimethylbenzene-purple); 123TMB – 1,2,3-Trimethylbenzene; 135TMB – 1,3,5-Trimethylbenzene; 124TMB – 1,2,4-Trimethylbenzene.

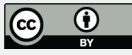

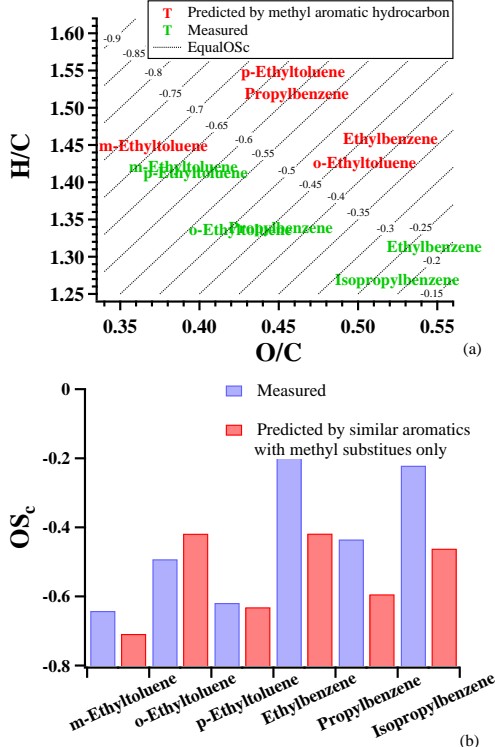



**ACPD**

doi:10.5194/acp-2015-871

**Impact of molecular structure on SOA formation**

L. Li et al.



**Figure 7.** Comparison of measured and predicted elemental ratio **(a)** and oxidation state **(b)** of SOA formed from longer alkyl substitute ($-C_2H_{2n+1}$, $n > 1$). Ethyltoluenes are predicted by corresponding xylenes and one substitute aromatic hydrocarbons are predicted by toluene. *Predicted elemental ratio of isopropylbenzene is same as propylbenzene (not showed in **a**).