# Peer review of "Impact of Molecular Structure on Secondary Organic Aerosol Formation from Aromatic Hydrocarbon Photooxidation under Low NOx Conditions"

_Atmospheric Chemistry and Physics, 2015_

## Referee Comment (RC1) · Anonymous Referee #1 · 11 Feb 2016

SUMMARY

This paper investigated the impact of molecular structure on the photooxidation of aromatic SOAs. The effect of the alkyl substitutes on the yield of chamber generated SOA, the chemical composition and the physical properties of SOA were studied. The authors concluded that oxidation of products promote the elevation of SOA yields. They found that the aromatic oxidation increased with increasing alkyl substitute chain length and it also varies with the branching position of an alkyl group on aromatic ring. Using their chamber data, the authors classified aromatics into five groups and ranked them as ortho (o-xylene and o ethyltoluene) > one substitute (ethylbenzene, propylbenzene and isopropylbenzene) > meta (m-xylene and m-ethyltoluene) > three sub-

stitute (trimethylbenzenes) > para (p-xylene and p-ethyltoluene). Overall, all data of this paper need statistical evaluation providing uncertainties in processed data or the errors associated with data points in Figures and Tables. The comparison of data from different aromatic systems and the data interpretation should be based on statistical significance. The explanation of SOA yields and the processed data from analytical instrument are very empirical and needs better interpretation with rationales based on kinetic mechanisms in both the gas phase and aerosol phase. The authors should provide better atmospheric imprecation of the observation and the outcomes of this study based on SOA formation in ambient environments and the classification of aromatic hydrocarbons in current air quality models .

Comments:

1. Page 6, line 5-10. The SOA formation has been performed at very dry conditions (RH<0.1%), which is very different from ambient environment. The SOA formation can be affected by humidity. Particularly, heterogeneous reactions is sensitive to aerosol water content because some of carbonyls and epoxides can be hydrated with available water in aerosol and oligomerized. Such reactions are also influenced by aerosol compositions and the hygroscopic properties of aerosol. In addition to the reactivity of oxidized carbons in aerosol, the aerosol phase water can modulates the reduction of viscosity of aerosol media, which also affect aerosol growth. Thus, the order in aerosol growth determined at very dry conditions may be/may not be different from aerosol growth in higher humidities. The authors should discuss about the potential influence humidity on aerosol growth and the rank observed in this study.

2. 2nd paragraph, Section 2.2. The aerosol samples were evaporated at 600oC followed by impaction. Such high temperature promotes charring of organic compounds and is able to modify chemical compositions of organic compounds. This should be clarified.

3. For Table 1, the author need to provide the uncertainty associated with SOA yields.

Please provide the unit with variables instead of footnote. All Figures and Tables need the uncertainties or the errors associated with data.

4. Section 3.1 SOA yield. (a) For SOA yields, the authors employed the two product model that was derived by Odum et al. The experimentally observed SOA yield varies with VOC/NOx ratios and NO/NO2 ratios. The authors describe that HC/NO ratios range 11.1 to 171. Does the aerosol at HC/NOx = 11 have the same chemical composition of the SOA at HC/NOx=171 ? Although all experimental conditions fall into the high NOx regions but they are various. Please discuss about the potential effect of NOx on SOA formation within the experimental conditions of this study. (b) The HC/NOx ratios between aromatic systems in Table are not same. Without the rationale for the effect of HC/NOx on SOA yields, the rank of SOA production from different SOA loses its meaning. (b) SOA yields are influenced by the amount of initial hydrocarbon and its reaction rate. When the aerosol is quickly formed, the loss of gaseous oxidized carbons to the reactor wall becomes smaller (Ng et al, 2007). The authors should clarify weather aerosol yields between different systems are not biased due to the potential loss of gaseous compounds to the wall by different reaction rates and initial experimental conditions. (c) In addition to the kinetic reactivity of hydrocarbons, the determination of SOA yields are subjective to the duration of chamber operation. The authors need to explain how the yield of SOA was determined based on reaction time and aerosol growth.

5. Page 8, lines 10-15 and Figure 1. It seems that the substitute length also affects the yield of SOA (C8 vs C9). The data should be treated by the statistical evaluation.

6. Figure 2. Where are the actual data point from each aromatic hydrocarbon in Figure 2 ? Please include the maker for each data point and the uncertainty of data points.

7. 1st paragraph, Section 3.2.1. It is difficult to follow the description of mass fragmentation in the text. It would be better to organize mass fragmentation information using Table. How does the aerosol have carboxylic functional groups (CO2). Does carboxylic

acid form via gas phase oxidaiton of hydrocarbons or autooxidaiton in particle phase ? What are the precursor structures to produce carboxylic groups.

8. Line 20-21, page 10. "While m/z 43 (C3H+7 ) and 57 (C4H+9 20 ) are often considered as markers for hydrogen-like organic aerosol . . .". What does "hydrogen-like organic aerosol" mean ? The explanation about mass fragmentation is unclear.

9. Last paragraph in section 3.2.1. The authors proposed that longer alkyl substitutes may not lower the oxidation per mass as further oxidation but observed the similar f44 and f43+57+71 of toluene suggesting unidentified oxidation. The interpretation of the analytical data is based on partitioning theory because authors' interpretation focused on gas phase oxidation. The oxidized carbons such as carbonyl are reactive in aerosol phase and can be transformed into oligomeric matter. In the past, the characterization of SOA have shown that oligomeric matter was significantly contributed to aromatic SOA mass. The interpretation of the data here was very empirical. The authors needs to rationale for observed data with better interpretation based on kinetic mechanisms.

10. Page 8, line 30 - Page 9, line 2, for Figure 1. It is uncertain whether the one alkyl substituent on the aromatic ring is clearly separated from meta-positioned aromatics without statistical significance. 11. Section 3.2.2 (H/C vs O/C). The data points in Figure S6 are scattered. The reviewer is sure how the authors chose the representative point from each diagram. The difference between the representative point from each diagram should be statistically determined. Without statistical assessment, it is hard to conclude the fact but it looks that difference in representative (averaged) points between systems would be insignificant within standard deviation of scattered data except few systems.

12. 2nd paragraph, in page 13. The explanation about H/C and O/C ratio is empirical and need better interpretation in the point of kinetical mechanisms.

13. Atmospheric Implication section. The authors need to provide the implication of the observation of this study to ambient environments. It has been known that toluene

is the most abundant in urban areas where NOx and humidity are high. Can the observation of this work be applicable to ambient environments? In the current CMAQ, there are two class of aromatics: high and low yield aromatics. Toluene is currently in the group of high yield aromatics. How can the observation of this paper be applied to the current air quality model ? Is the classification of five groups of aromatics in this study meaning ? All chamber studies have been limited to high concentration of initial conditions (VOC and NOx) due to detection limit of analytical instrument. What is the implication of this work to the SOA in the low concentration environments (ambient air) ?

14. For Figure 5, the author could explain why the predicted density of 1,3,5-TMB is much lower than the measured density.

15. For Figure 6, the author should provide the uncertainty of VFR(end) values.

16. In the Table 1, the author should check the M0 value for 1215A, which is very high (M0=1501).

---

## Referee Comment (RC2) · Anonymous Referee #2 · 1 Apr 2016

The manuscript presents information on yields for the photooxidation of single ringed-aromatic structures. The senior author has been measuring yields from aromatic hydrocarbons (AHCs) for more than 15 years and is well versed. In this particular manuscript, the structure of the aromatic hydrocarbon (AHC) has been varied to examine differences in the organic aerosol (OA) yield. Thus, a series of 12 alkyl-substituted C8 and C9 AHCs have been examined. For these experiment, the aerosol yield has been determined using the Odum two-product model. Other OA parameters examined include the ratio of aerosol mass spectrometry (AMS) peaks attributed to OA, the OA oxygen-to-carbon (O: C) ratio, the oxidation state (OS), density, and volatility. The authors conclude that changes in the OA chemical composition and volatility influences

the yield typically by increasing the mass for increased oxidation. The authors also consider the atmospheric implications of this study.

The study addresses an issue of perhaps abstruse importance. The oxidation of alkyl-substituted AHCs and the formation of secondary organic aerosol (SOA) has been examined extensively over the last 20 years and this work appears to cover some old territory. Many of these topics were addressed in Odum et al. 1997a, b (authors' references) and the present manuscript provides a bit more insight. Admittedly, the AMS was not around and the aerosol density from the volume distribution was considered to be unity in the 1990s. However, this work also represents a step backwards. Whereas Odum et al. 1996, 1997a sought to simplify aerosol yields, this work goes in the other direction and makes an argument (at least implicitly) that the yields should be addressed more precisely, a contention that I don't feel has been justified (see Table 2). That said, the experiments appear to have been carefully performed and there certainly are enough of them. Unfortunately, the initial conditions are all over the map and makes it very difficult to get a sense of the reproducibility of a given experiment. Virtually, nothing is said about uncertainty.

My major substantive comments regarding the manuscript are as follows: (1) these experiments hardly qualify as being under low NOx conditions. The removal of RO2 radicals competitively by NO or RO2 determines the regime that the reaction is in. I would consider the low NOx regime as conditions where the RO2 radical-radical reactions become more important than the NO reaction. As a rule of thumb, I would say that this is certainly at no more than 5 ppb of NO for the conditions of these experiments depending, of course, on the specific RO2 radicals from the precursor AHC. (2) The version of SAPRC referenced to Carter and Heo (2013) is specifically geared to ozone prediction, that is, conditions where NOx dominates early product generations (two, at most three). As stated in Carter and Heo, (2013; Atmospheric Environment) SAPRC-11 is not geared for PM predictions. For this to be the case, the importance of RO2 + RO2 reactions should be adequately predicted as should the SOA mass. That

is the point of the model and to predict radical concentrations as they were an end in themselves. The experimental SOA values could then constrain the model. That said, I question how well SAPRC-11 predicts radical concentrations under low NOx conditions. (3) Several sections need a complete writing overhaul. Section 3.2.1 is barely comprehensible. Many sections in the Results and Discussion presents data (e.g., S6) as if they were self-interpretive. The manuscript is written for the audience being other scientists in their research group. There is considerable jargon and the writing is highly imprecise hardly worthy of a scholarly journal. (4) To the extent possible, the authors should give a greater physical interpretation of the metrics they present in Section 3. Some are obvious (e.g., Sec. 3.2.2) others far less so (Sec. 3.2.1; 3.4).

Some comments and suggestions:

P5, L27. The goal of the research states the obvious. Perhaps more insight will motivate the reader to actually read the paper.

P6, L7. UV-350 bulbs have considerable radiation in the UVB which accentuates the photolysis of carbonyl compounds to a considerable extent which accelerates PM formation by increasing the radical concentrations. Thus, the two-product parameters developed (Table 2) may not be applicable for predictions of ambient AHC PM (Tables 2 and S3)

P6, L18. Provide a chemical name for the standard OEKANAL.

P6, L23. How is mixing achieved in this large chamber; fan, diffusion, other?

P7, L10. Calling a bunch of peaks attributed to innumerable organic compounds completely fragmented by 70 eV electrons as a chemical composition stretches the concept of molecules beyond recognition. I would hardly call this metric a chemical composition in any traditional sense. Perhaps the word "effective" could be incorporated to indicate that this is simply a parameterized metric.

P7, L23. The authors should reference the 2013 Atmospheric Environment article by

Carter and Heo rather than the CARB report. The article went through peer-review and should be more reliable.

P8, L11. The sentence is unneeded; include the information in the caption.

P9, L3. To use the word "claimed" in a pejorative fashion is particularly bad form. If you believe the statement in Odum et al. is wrong, simply state it.

P9, L14. Delete the sentence. The supplement does nothing to support the sentence other than to simply repeat itself and refer to a paper in preparation. Nothing is gain by including the sentence in the paper or in the supplement.

P9, L23. Why is the assumption needed? The two-product model is just a fitting exercise anyway.

P9. It might be worthwhile to examine partitioning using a volatility basis set (VBS) to see if any insight could be gained beyond the standard two-product fit which at this point is rather dated. This might provide a more useful metric for describing the partitioning of the AHC products.

Sec 3.2.1. This section suffers from a lack of an understandable interpretation of the various fragments from the AMS output and their combinations into the combined metrics (e.g. Eq 1). An annoying aspect of this section is the comparison with other work before any interpretation is provided (e.g., P11, L5, 16; P12 L6, 25,.....). How do we even know that the conditions are applicable between these experiments and the ones being compared to?

P13, L7. LV-OOA and SV-OOA are presented both undefined and without context.

P13, L8. The sentence as written belongs in the introduction. The intent of the sentence needs a rationale from the data. It is not self-evident.

P14, L21. "a more accurate metric...." More accurate than what?

P16, L1. Eliminate first name for Borrás.

P17, L15. Use of the term "theory" (in any scientific sense) strikes me as somewhat pretentious. I would consider it as more of a conjecture.

P18, L8-16. What does the term "extremely low" mean? Provide a value for comparison. I would characterize most of this part as speculation. Also, experimental limitations in Forstner et al., 1997 (their specific quartz filter configuration) suggested that the furan-type compounds were in the particle phase but were almost certainly in the gas phase. The metrics in the present paper refer only to particle phase OA. The reference should be avoided here.

Section 5. I seriously doubt that any of these parameterizations would appear in any wide-used air quality model. I would consider the work mainly for academic purposes.

P19, L2. Replace "entend" with "extent".

P19, L3. Nothing is "proved" here. The work simply provides "evidence for".

P25, L8. Replace the ACPD manuscript with newly published ACP paper.

Table 1, S2 and text. How many AHCs were studied: the text say twelve AHCs (P5, L7), Table 1 gives ten AHCs, and Tables 1 and S2 together give fourteen. Which is the right number? (For good measure, Figure 6 shows eleven precursors.)

Table 3. What's the point of the table if the p value are greater than 0.05. Certainly, the p-value for VFR and k (OH) is not zero.

Figure 1. The final points control the shape of the curve. The data below 20 ug m-3 would be of most interest for atmospheric applications.

Figures 2, 3, 7. Am I missing something? Why are there no symbols for the values; is there that much uncertainty? The use of colors for the precursors in Table 2 is particularly annoying. Are not words sufficient without colors?

In Figure 4, what are the estimated uncertainties in the model-generate radical concentrations. Carter and Heo, 2013 suggests that these could be substantial.

Table S3. How about the C9-trimethyl compounds studied? Why not put the parameters obtained from the TMB compounds in the table? It would also be informative to include the data for toluene (which must have been studied at some point) for comparison with the other single position substituents, ethylbenzene and n-,i-propylbenzene.

Table S4. The value of this table for predicting radical concentrations is very limited as noted above. It may be useful for urban NOx conditions but not where RO2 + RO2 is the dominant source of the aerosol. It doesn't surprise me that the p-values for virtually all comparisons in S5 are no different than the null hypothesis.

SI Table 3 should be Table S6. Use lower case k in the table.

---

## Referee Comment (RC3) · Anonymous Referee #3 · 28 Apr 2016

In this manuscript, the authors Li et al. studied formation of secondary organic aerosol from aromatic hydrocarbons, quantified the formation yields and investigated the chemical and physical properties in order to infer the differences in underlying mechanisms between different molecular structures. The hydrocarbons studied spanned from C7 to C9 and from 1 to 3 alkyl substitutes on the aromatic ring. These hydrocarbons are important for urban areas, and have not been well studied compared to benzene and toluene. The authors presented a variety of measurements (yields, aerosol mass spectrometer, density, volatility) conducted with state of the art instrumentation. The experiments are well designed and the manuscript is generally well written with relatively minor typos. However, I believe that the data interpretation is weak and it is very

difficult to draw conclusions based on the data presented, especially with regards to SOA yields. I recommend major revisions before the manuscript can be considered for publication in ACP.

Major comments:

1. The biggest weakness of this manuscript is that the SOA yields are quite varied and many of the conclusions drawn by the authors are not very convincing. In Fig. 1, there are no error bars shown for the individual yields and Mo, and for the empirical fits. For example, the authors conclude from this figure that ortho compounds yields are similar and regression was performed on the combined data set. However, one could easily argue that o-xylene yields and m-ethyltoluene yields are similar and should be fitted to the same curve. Also, one of the major conclusions about ortho-compounds having higher yields is very weak. At atmospherically relevant Mo (Mo < 40 ug/m3), the yields are all very similar. Any differences in mechanisms do not translate to significant differences in formation yields (smaller than experimental uncertainties in yields). Therefore, one can argue that molecular structure plays a relatively minor role.

2. Another critique I have about this manuscript is the lack of mechanistic insights revealed by the experiments. I read the earlier paper on carbon dilution theory by the same authors, and found that the AMS data were very useful in identifying the general effect: methyl substituents divert OH oxidation from the ring, leading to less oxidized SOA components. A similar approach is used here but much less effectively. It is unclear from the proposed theory why the position of the alkyl substituents matters. Again, the authors claim that SOA from ortho-compounds is more oxidized than that from para- and meta-compounds. Why is that the case? The authors can make this manuscript much stronger if they can propose mechanisms along with careful experimental work to support them. Since the theme of the paper is molecular structure, I think these explanations are very important and deserve more attention. Can the authors propose any mechanisms (does not have to 100% proven, only needs to be plausible) that may be consistent with the results? AMS data are not suitable for resolv-

ing isomer-specific differences. More speciated measurements (such as LC or GC/MS) will provide more insights.

3. Similar to my previous comment (and other reviewers' comments), the AMS does not really provide information about specific chemical composition. For example, the statement in Section 3.2.2 "SOA components from all isomers are located in between slope= $-1$ and slope= $-2$ lines suggesting that SOA from these aromatic hydrocarbons is composed primarily of acid (carbonyl acid and hydroxycarbonyl) and carbonyl (ketone or aldehyde) like functional groups". The elemental ratios does not suggest that the SOA *contain* these functional groups, but rather they suggest that the SOA composition *evolve* as if they were adding these function groups. This is a very important difference that should be noted throughout the manuscript. The AMS elemental ratios help elucidate the bulk composition and evolution, not the specific chemical composition and mechanism, as suggested by the manuscript. I believe that AMS was useful in showing O/C changes from benzene to toluene to xylenes, but the specific isomeric differences in this paper are not convincingly argued in this manuscript.

4. It is not clear to me why the authors chose to conduct the experiments under low-NOx conditions. These highly substituted aromatic hydrocarbons are emitted in urban areas, and have lifetimes around 1 day or so. Even in rural areas, the NO levels are quite high, resulting in at least 50% oxidation by RO2+NO pathway (see Ortega et al., ACP, 2014). One could argue that the experiments conducted are still experiments with NOx (unlike HO2 dominated experiments). However, the HC/NOx ratios are not fixed (ranges over 1 order of magnitude), bringing into question the relative role of NOx. Why aren't the experiments conducted with a fixed HC/NOx ratio, or with NOx so high that it is not limiting?

Other comments:

Pg 5 line 2: what is the significance of the furanone relative abundances? Does it point to a particular propensity of fragmentation and/or SOA formation from one particular

molecular structure?

Pg 9 line 20: The parameters are for fitting purposes only, and are semi-empirical at best. Because the Odum 2-product equations are non-linear, alpha and K are coupled parameters (i.e. one can use a higher alpha and lower K and still get a decent fit to the experimental data). Therefore I suggest the authors not derive insights into relative volatilities from the fitted parameters.

Pg 14 line 2-11: Is it possible that photolysis of aromatic carbonyls is playing a role in affecting the H/C of SOA?

Table 3: It seems to me that the correlations are quite weak (around 0.5 at best) and the p-value is always greater than 0.05, sometimes much greater. From a statistics point of view, the correlations are inconclusive. I suggest rewording in main text to call these "weakly correlated". (". . . best correlated. . ." is misleading).

Sect 3.3.2: Can the authors isolate the effect of kinetics and molecular structure by comparing VFR at the same extent of reaction?

The authors alluded to the carbon dilution theory they proposed in an earlier publication (Li et al., 2015a) many times throughout the manuscript without explaining the theory. This theory is relatively novel and is not well known. Therefore I suggest adding a short section to explain the theory early on in the manuscript and show how this will be applied to the molecular structures studied in this work.

Technical comments:

Pg 2 Line 5: eight to nine carbon should be "C8- to C9-" or "eight- to nine-carbon"

Pg 2 Line 23 IPPC should be IPCC

Pg 2 Line 26 What does growth potential mean?

Pg 3 Line 14 Toluene and C8 aromatics do not dominate aromatic SOA, because they are not SOA themselves, but are precursors to SOA. I suggest rewording this sentence.

Pg 5 line 12: references to mass loading (Shilling et al. 2009 and Pfaffenberger et al 2013) and NO effect (Eddingsaas et al. 2012) on SOA formation are for a-pinene, not for aromatics. The authors should clarify that point.

Pg 6 line 13-20: are the chemicals used without further purification?

Pg 7 line 8-9: "Volume fraction remaining" should be all capitalized

Pg 7 line 13-14: clarify that fx is the *mass* fraction of organic signal at m/z = x

Pg 7 line 19: "squirrel" should be capitalized

Pg 8 line 10: organic mass concentration should be Mo (o for organics), not M0. M0 would suggest that it is an initial mass concentration

Pg 10 line 17: what does the n stand for? It might be more useful to use n to denote the carbon number of the alkyl substitute. So for m/z 57 is derived from an ethyl-substituted aromatic, so n would be 2, and the formula would be $C_{(n+1)}H_{(2n+1)}O+$.

Pg 10 line 21: "hydrogen-like organic aerosol" should be "hydrocarbon-like organic aerosol"; it seems quite obvious to me that it is not possible to have a $C_3H_7$ or $C_4H_9$ fragment from aromatic compounds here. Even isopropyl benzene does not yield $C_3H_7$ upon EI fragmentation (see NIST spectra). I suggest removing that explanation to make it more concise.

Pg 11 line 25: This section is very hard to follow. Is there a proposed mechanism of how an oxidation product of aromatic compounds can produce m/z 43 that is also consistent with proposed carbon dilution theory? It would be very beneficial here to use a figure to illustrate the key points.

Pg 12 line 1: do the authors mean ROOH here? The bicyclic peroxides formed in aromatic oxidation are internally bridged, and are therefore ROOR, not ROOH (or hydroperoxides). I presume that in the presence of NOx, RO2+HO2 is negligible and ROOH is not formed. Also, is there a reference for peroxides not yielding $CO_2+$ from

the AMS literature?

Pg 13 line 8: "The current study concentrated on" should be "The current study concentrates on"

Pg 17 line 20: the chemical formula C2H2n+1 seems wrong

Figure 2: there should not be a continuous scale for molecular structure; Also for all the figures, the experiment numbers should be noted as such. Otherwise the numbers do not mean anything to readers

Figure 4: can the bars be color coded to correspond to those in other figures (e.g. Figs. 5 and 6)?

---

## Author Comment (AC1) · 5 May 2016

The statistical evaluation is addressed in the revised manuscript and a detailed analysis of the statistical evaluation are included in the replies to Comment 5, 10 and 11. Additional mechanism discussion is included in the revised manuscript to provide interpretation for the results. See replies to Comment 9 and 12 for details. The implications of this work to ambient environments and air quality models is further enhanced as described in Comment 13. A point by point reply to the reviewer is listed below.

Please also note the supplement to this comment:

[Figure]

http://www.atmos-chem-phys-discuss.net/acp-2015-871/acp-2015-871-AC1-supplement.pdf

[Figure]

**Supplement:**

*Referee's Comment in Italic Font;* Author's response in Red and Manuscript revision in Blue without Italic Font.

**Reviewer #1**

*This paper investigated the impact of molecular structure on the photooxidation of aromatic SOAs. The effect of the alkyl substitutes on the yield of chamber generated SOA, the chemical composition and the physical properties of SOA were studied. The authors concluded that oxidation of products promote the elevation of SOA yields. They found that the aromatic oxidation increased with increasing alkyl substitute chain length and it also varies with the branching position of an alkyl group on aromatic ring. Using their chamber data, the authors classified aromatics into five groups and ranked them as ortho (o-xylene and o ethyltoluene) > one substitute (ethylbenzene, propylbenzene and isopropylbenzene) > meta (m-xylene and m-ethyltoluene) > three substitute (trimethylbenzenes) > para (p-xylene and p-ethyltoluene). Overall, all data of this paper need statistical evaluation providing uncertainties in processed data or the errors associated with data points in Figures and Tables. The comparison of data from different aromatic systems and the data interpretation should be based on statistical significance. The explanation of SOA yields and the processed data from analytical instrument are very empirical and needs better interpretation with rationales based on kinetic mechanisms in both the gas phase and aerosol phase. The authors should provide better atmospheric imprecation of the observation and the outcomes of this study based on SOA formation in ambient environments and the classification of aromatic hydrocarbons in current air quality models .*

The statistical evaluation is addressed in the revised manuscript and a detailed analysis of the statistical evaluation are included in the replies to Comment 5, 10 and 11.
Additional mechanism discussion is included in the revised manuscript to provide interpretation for the results. See replies to Comment 9 and 12 for details.
The implications of this work to ambient environments and air quality models is further enhanced as described in Comment 13.
A point by point reply to the reviewer is listed below.

*Comments:*
*1. Page 6, line 5-10. The SOA formation has been performed at very dry conditions (RH<0.1%), which is very different from ambient environment. The SOA formation can be affected by humidity. Particularly, heterogeneous reactions is sensitive to aerosol water content because some of carbonyls and epoxides can be hydrated with available water in aerosol and oligomerized. Such reactions are also influenced by aerosol compositions and the hygroscopic properties of aerosol. In addition to the reactivity of oxidized carbons in aerosol, the aerosol phase water can modulates the reduction of viscosity of aerosol media, which also affect aerosol growth. Thus, the order in aerosol growth determined at very dry conditions may be/may not be different from aerosol growth in higher humidities. The authors should discuss*

*about the potential influence humidity on aerosol growth and the rank observed in this study.*

We agree SOA formation is potentially affected by humidity due to heterogeneous reaction. The current study provides a fundamental relationship among SOA formed from different aromatic isomers under dry conditions.

We add the following sentences in Page 19, Line 7 (end):
Previous studies found that the humidity insignificantly impacts SOA yield from aromatic hydrocarbons (Cocker, et al., 2001) or maintains the SOA yield relationship between isomers (Zhou, et al., 2001). Therefore, it is predicted that the observation found under dry conditions in this study, especially the molecular structure impact on SOA formation from different aromatic isomers could be extended to atmospherically relevant humidity conditions. However, recent studies observe that the hydration of carbonyls and epoxides could lead to further heterogeneous reaction and oligomerization (Jang, et al., 2002; Liggo, et al., 2005; Minerath and Elrod, et al., 2009; Lal, et al., 2012). It is possible that aerosol compositions and the hygroscopic properties could be altered after the heterogeneous reactions, especially under humid conditions. The impact of molecular structure impact on SOA formation under humidity condition needs to be further studied to extend the findings in the current work.

*2. 2nd paragraph, Section 2.2. The aerosol samples were evaporated at 600oC followed by impaction. Such high temperature promotes charring of organic compounds and is able to modify chemical compositions of organic compounds. This should be clarified.*

The AMS used in our study followed standard measurement techniques for SOA studies. Heating in the HR-TOF-AMS is conducted under vacuum, which minimizes "charring" of the organic species.   We add a "under vacuum" at page 7 line 12 (2nd paragraph, Section 2.2) for clarification. The sentence now reads
 "The sample was vaporized by a 600 $^{\circ}$C oven under vacuum followed by a 70 eV electron impact ionization."

*3. For Table 1, the author need to provide the uncertainty associated with SOA yields. Please provide the unit with variables instead of footnote. All Figures and Tables need the uncertainties or the errors associated with data.*

We insert following sentence addressing SOA uncertainty in the methods section at Page 8 Line 9: "The uncertainty associated with 10 replicate m-xylene and NO experiments SOA yield is <6.65%."

The standard deviations of O/C, H/C, $f_{44}$, $f'_{43+57+71}$, $OS_c$ and $VFR_{end}$ are now included in the revised manuscript along with the standard deviations shown in the original manuscript for density.

We modified the manuscript to provide the unit with variables instead of footnotes as requested.

*4. Section 3.1 SOA yield. (a) For SOA yields, the authors employed the two product model that was derived by Odum et al. The experimentally observed SOA yield varies with VOC/NOx ratios and NO/NO2 ratios. The authors describe that HC/NO ratios range 11.1 to 171. Does the aerosol at HC/NOx = 11 have the same chemical composition of the SOA at HC/NOx=171 ? Although all experimental conditions fall into the high NOx regions but they are various. Please discuss about the potential effect of NOx on SOA formation within the experimental conditions of this study. (b) The HC/NOx ratios between aromatic systems in Table are not same. Without the rationale for the effect of HC/NOx on SOA yields, the rank of SOA production from different SOA loses its meaning. (b) SOA yields are influenced by the amount of initial hydrocarbon and its reaction rate. When the aerosol is quickly formed, the loss of gaseous oxidized carbons to the reactor wall becomes smaller (Ng et al, 2007). The authors should clarify weather aerosol yields between different systems are not biased due to the potential loss of gaseous compounds to the wall by different reaction rates and initial experimental conditions. (c) In addition to the kinetic reactivity of hydrocarbons, the determination of SOA yields are subjective to the duration of chamber operation. The authors need to explain how the yield of SOA was determined based on reaction time and aerosol growth.*

Only ~10% of experiments have HC/NO larger than 60 ppbC:ppb. The majority of experiments have similar NO conditions. NO could impact SOA formation by either forming organic nitrate or reducing peroxide radical concentrations. First, aerosol at HC/NO = 11 ppbC:ppb may have slightly different SOA chemical composition at HC/NO$_x$=171 ppbC:ppb due to a higher organic nitrate fraction. However, less than 5% of organic matter is organic nitrate in this study and therefore exerts little impact on overall yield and chemical composition. Second, lower HC/NO might lead to lower peroxide radical concentrations and therefore lower SOA yield. However, no significant correlations between SOA yield and radical concentrations are observed as discussed at Page 9 Line 11-15 and presented in Table S5. This shows the lack of correlation between radical concentration and yield. Therefore, NO is not a major factor to SOA yield within the experimental HC:NO range used in this study.

Seeded experiments to minimize gaseous compounds wall loss were conducted in our chamber experiment with no significant difference observed between the seeded and non-seeded experiment. This indicates that the gas-phase wall might not be expected to be significant for experiments in our chamber for this aromatic SOA study.
We add the following sentence at line 8 Page 6in the revised manuscript.
Seeded experiments to minimize wall effects have also been conducted in our chamber experiment with no measurable difference observed between the seeded and non-seeded experiment.

We agree that SOA yields are subjective to the duration of chamber operation. In this work, SOA yield is calculated after 6-8 hours of photooxidation for each experiment. All the precursors studied have similar k$_{OH}$ (Table S1) or similar k$_{OH}$[Precursor] (Li, et al., 2016). We selected comparable photooxidation time (6-8 hours) for all precursors and therefore the SOA yields are comparable.

*5. Page 8, lines 10-15 and Figure 1. It seems that the substitute length also affects the yield of SOA (C8 vs C9). The data should be treated by the statistical evaluation.*

We agree with the reviewer that substitute length also affects the yield of SOA (C8 vs C9) as stated on Page 10, line 1-7. However, the difference caused by substitute length is less significant than substitute location as shown in Fig. 1. Differences in SOA yield due to length is within the SOA yield standard deviation (now shown). There may be differences among all one substitute aromatics, which is answered in detail in the reply to Comment 10. We add the following sentence at line 6 Page 10 in the revised manuscript. "However, the differences between xylenes and their corresponding ethyltoluenes are not statistically significant."

Section 4 further discusses the differences associated with substitute length (original manuscript on Page 10, line 7.): "These differences are explained by the proposed alkyl group dilution effect (Sect. 4)."

*6. Figure 2. Where are the actual data point from each aromatic hydrocarbon in Figure2 ? Please include the maker for each data point and the uncertainty of data points.*

The actual data points from each aromatic hydrocarbon in Figure 2 become obvious after adding standard deviation on both the x-axis and the y-axis as suggested in Comment 3.

*7. 1st paragraph, Section 3.2.1. It is difficult to follow the description of mass fragmentation in the text. It would be better to organize mass fragmentation information using Table. How does the aerosol have carboxylic functional groups (CO2). Does carboxylic acid form via gas phase oxidation of hydrocarbons or autooxidaiton in particle phase? What are the precursor structures to produce carboxylic groups.*

We add Table S6 to provide the requested peak information.

$CO_2^+$ is a common peak from AMS when measuring aerosol. It is less likely that $CO_2^+$ is associated with carboxylic functional group in this study. There are other possible $CO_2^+$ fragment sources. For example $CO_2^+$ may come from the oligomerization of small cyclic furanones described in earlier work (Li, et al, 2016).

*8. Line 20-21, page 10. "While m/z 43 (C3H+7 ) and 57 (C4H+9 20 ) are often considered as markers for hydrogen-like organic aerosol . . .". What does "hydrogen-like organic aerosol" mean ? The explanation about mass fragmentation is unclear.*

This is a typo. It should read "hydrocarbon-like organic aerosol" instead of "hydrogen-like organic aerosol". An m/z 57 and m/z 71 in field studies or ambient atmosphere usually associate with hydrocarbon-like organic fragments, $C_4H_9^+$ and $C_5H_{11}^+$, respectively (Zhang, et al., 2005; Ng, et al., 2010). However, m/z 57 and m/z 71 in chamber studies, especially for ethyl and propyl substituted aromatics photooxidation, are majorly $C_3H_5O^+$ and $C_4H_7O^+$, which are oxygenated organic aerosol (OOA). Ng. et al (2010) developed $f_{43}$ (majorly $C_2H_3O^+$) and $f_{44}$ ($CO_2^+$) for ambient OOA categorization without m/z 57 and m/z 71 since m/z 57 and m/z 71

are majorly HOA in ambient atmosphere. However, $C_3H_5O^+$ (m/z 57) and $C_4H_7O^+$ (m/z 71) should also be included beside $C_2H_3O^+$ in SOA chamber studies as OOA to compare the oxidation of different aromatic hydrocarbons. We replace Line 20-24, page 10 with sentences below:

While m/z 57 ($C_4H_9^+$) and m/z 71 ($C_5H_{11}^+$) are often considered as markers for hydrocarbon-like organic aerosol in ambient studies (Zhang et al., 2015; Ng et al., 2010), oxygenated organic aerosol C3H5O+and C4H7O+ are the major fragments at m/z 57 and m/z 71, respectively, (Fig. S4, Table S6) in current chamber SOA studies, especially during the photooxidation of ethyl and propyl substituted aromatics. Therefore, m/z 57 and m/z 71 are also considered beside $C_2H_3O^+$ at m/z 43 in SOA chamber studies as OOA to compare the oxidation of different aromatic hydrocarbons.

*9. Last paragraph in section 3.2.1. The authors proposed that longer alkyl substitutes may not lower the oxidation per mass as further oxidation but observed the similar f44 and f43+57+71 of toluene suggesting unidentified oxidation. The interpretation of the analytical data is based on partitioning theory because authors' interpretation focused on gas phase oxidation. The oxidized carbons such as carbonyl are reactive in aerosol phase and can be transformed into oligomeric matter. In the past, the characterization of SOA have shown that oligomeric matter was significantly contributed to aromatic SOA mass. The interpretation of the data here was very empirical. The authors needs to rationale for observed data with better interpretation based on kinetic mechanisms..*

We suggest that longer alkyl substitutes may not lower the oxidation per mass by relying on both the observation in section 3.2.1 and the elemental ratio prediction method (alkyl dilution theory) in Discussion part (section 4). Higher OSc (lower H/C and higher O/C) is observed in longer chain single alkyl substitute aromatics than the alkyl dilution theory predicts from toluene data. Oligomerization, as suggested by the reviwer, consists of highly oxidized monomers (e.g. glyoxal) and could therefore also increase the overall OSc. Therefore, we add following sentence at Page 12 Line 17:
It is also possible that oligomerization from highly oxidized carbonyls contribute more to the SOA formation from aromatics with long chain alkyl substitute.

Previous theoretical studies predict that alkyl substitute plays a role in oligomerization, e.g. glyoxal favors acetal oligomerization and methylglyoxal prefers aldol condensation to form oligmers (Barsanti and Pankow, 2005; Krizner, et al., 2009). However, these two oligomerization kinetic mechanisms produces products with similar formulas. No matter which mechanism is favored, SOA elemental ratio should not be affected. Therefore, the percentage of oligomerization to other reaction mechanisms rather than the difference in oligomerization mechanism seems a better explanation for the observation in SOA formed from long chain single alkyl substitute aromatics.
Therefore, we also add the following sentences in the Discussion section at Page 18 Line 6:
It is also possible that oligomerization from highly oxidized carbonyl component might be more favored for long chain single alkyl substituted aromatics.

*10. Page 8, line 30 - Page 9, line 2, for Figure 1. It is uncertain whether the one alkyl substituent on the aromatic ring is clearly separated from meta-positioned aromatics without statistical significance.*

We agree that one alkyl substituent on the aromatic ring SOA yield is not clearly separated from meta-positioned aromatics in Figure 1. However, it can be concluded from chemical composition and volatilities (VFR) that one alkyl substitute aromatic hydrocarbon is more oxidized than meta-positioned aromatics. The statistical parameter for the curve fitting is provided in the revised Table 2 as mentioned in the reply to Comment #5. It can be clearly seen that MSRE is much larger in one substitute aromatic fitting than meta-aromatic fitting. This indicates potential SOA yield differences within the one substitute aromatic hydrocarbons along with observed SOA chemical composition differences among one substitute aromatic hydrocarbons (Sections 3.3.2 and 3.2.3). Further studies are warranted to provide more information on SOA yield. At this time, we fit one alkyl substitute aromatics and meta-position aromatics separately.

*11. Section 3.2.2 (H/C vs O/C). The data points in Figure S6 are scattered. The reviewer is sure how the authors chose the representative point from each diagram. The difference between the representative point from each diagram should be statistically determined. Without statistical assessment, it is hard to conclude the fact but it looks that difference in representative (averaged) points between systems would be insignificant within standard deviation of scattered data except few systems.*

The scattering in Fig S6 is majorly due to a few early period data at the low mass loading when aerosol just starts to form, which have higher H/C and lower O/C than the latter time. However, *o*-xylene data contains data that does not follow the aerosol aging trend observed for the other isomers. Therefore, we delete the obvious outliners in Fig S6-f-*o*-xylene and Fig S6-6 o-Ethyltoluene and adjust Fig. 3, Fig. 4 and Fig 7 accordingly. We provide standard deviation for H/C, O/C and OS$_c$ in Fig 4, Fig 7b and supplemental materials as answered in question 3.

*12. 2nd paragraph, in page 13. The explanation about H/C and O/C ratio is empirical and need better interpretation in the point of kinetical mechanisms.*

The kinetic parameters (k$_{OH}$) for initial oxidation of ortho, meta and para containing aromatic hydrocarbons are all similar (Table S1). This suggests there may not be a significant difference among all these isomers from kinetic perspective. We draw the conclusion in the paper empirically based on what we observed in our measurement.

*13. Atmospheric Implication section. The authors need to provide the implication of the observation of this study to ambient environments. It has been known that toluene is the most abundant in urban areas where NOx and humidity are high. Can the observation of this work be applicable to ambient environments? In the current CMAQ, there are two class of aromatics: high and low yield aromatics. Toluene is currently in the group of high yield aromatics. How can the observation of this paper be applied to the current air quality model? Is the classification of five groups of aromatics in this study meaning? All chamber studies have been limited to high concentration of initial conditions (VOC and NOx) due to detection limit of*

*analytical instrument. What is the implication of this work to the SOA in the low concentration environments (ambient air)?*

We address the relevance of $NO_x$ conditions in current study to the ambient atmosphere at the beginning of Atmospheric Implication section. Toluene remains the second highest SOA yield (lower than benzene, especially at higher mass loading, see Li, et al., 2016) precursor according to our earlier work (Li, et al., 2016); however, it is not a target aromatic in current studies. This paper focuses on isomer or molecular structure impact on SOA formation. Current work provides sufficient data to distinguish among para, meta and ortho position containing aromatics and therefore is able to subcategorize the previous "low" and "high" yield aromatics (e.g. only one p-xylene data point in Odum, et al., 1997). The five groups of aromatics and their two product modeling curve fitting provide the practical parameter for more detailed SOA modeling. All SOA yield data provided is under more atmospherically relevant $NO_x$ conditions than the earlier "high" and "low" yield work improving reliability of fit parameters as inputs to atmospheric models.    Study at the lower atmospheric NOx concentrations provides yields twice as high as those from earlier work at very high NOx concentrations greatly impacting the model predictions from model prediction (e.g., CMAQ).

We add following sentences to emphasize the importance of this study to model and ambient environment at Page 19, Line 7:
Moreover, the five subcategories of aromatics and their two product modeling curve fitting parameters in this work at more realistic $NO_x$ loadings provide a more precise prediction of SOA formation form aromatic hydrocarbons under atmospheric conditions.

*14. For Figure 5, the author could explain why the predicted density of 1,3,5-TMB is much lower than the measured density.*
Thank you for this observation. In fact we observe density underestimation in all meta position containing aromatic hydrocarbons including *m*-xylene, *m*-ethyltoluene and *1,3,5*-TMB. The underestimation is associated with a bias in elemental ratio analysis from AMS as discussed in Li, et al., 2016; Nakao, et al., 2013.

We add following sentences in Page 16 Line 13:
A comparatively large negative error is found in meta containing aromatic hydrocarbons including *m*-xylene, *m*-ethyltoluene and *1,3,5*-trimethylbenzene. It is noted that there should be more alkyl substitutes in SOA formed from meta position aromatics than other aromatics since meta position alkyl substitutes are more likely to participate into SOA products than other aromatics (Section 3.2.1 and Section 3.2.2). Previous work suggests that the increase of methyl groups could lead to a change in several key organic fragments (e.g., $CO^+$, $CO_2^+$ and $H_2O^+$) thereby altering the default fragment table for elemental ratio analysis. This agrees with the density underestimation in SOA formed from meta position aromatics and supports the preference of meta position alkyl substitute to SOA products.

*15. For Figure 6, the author should provide the uncertainty of VFR(end) values.*
We add uncertainty of $VFR_{(end)}$ values in revised manuscript

*16. In the Table 1, the author should check the M0 value for 1215A, which is very high (M0=1501).*

It should be 151. Fixed.

***Reference***

Jang, M., Czoschke, N. M., Lee, S., and Kamens, R. M.: Heterogeneous atmospheric aerosol production by acid-catalyzed particle-phase reactions, Science, 298(5594), 814-817, doi:10.1126/science.1075798, 2002.

Krizner, H. E., De Haan, D. O., and Kua, J.: Thermodynamics and Kinetics of Methylglyoxal Dimer Formation: A Computational Study, J. Phys. Chem. A, 113(25), 6994–7001, doi: 10.1021/jp903213k, 2009.

Lal, V., Khalizov, A. F., Lin, Y., Galvan, M. D., Connell, B. T., and Zhang, R.: Heterogeneous reactions of epoxides in acidic media. J. Phys. Chem. A., 116(24), 6078-6090, doi: 10.1021/jp2112704. 2012.

Li, L., Tang, P., Nakao, S., Chen, C.-L., and Cocker III, D. R.: Role of methyl group number on SOA formation from aromatic hydrocarbons photooxidation under low-$NO_x$ conditions, Atmos. Chem. Phys., 16, 2255–2272, doi:10.5194/acp-16-2255-2016, 2016.

Liggio, J., Li, S.-M., and McLaren R.: Heterogeneous reactions of glyoxal on particulate matter: Identification of acetals and sulfate esters, Environ. Sci. Technol. 39(6): 1532-1541, dio: 10.1021/es048375y, 2005.

Nakao, S., Tang, P., Tang, X., Clark, C. H., Qi, L., Seo, E., Asa-Awuku, A. and Cocker III, D. R.: Density and elemental ratios of secondary organic aerosol: Application of a density prediction method, Atmos. Environ., 68, 273-277,doi:10.1016/j.atmosenv.2012.11.006, 2013

Minerath, E. C. and Elrod, M. J.: Assessing the potential for diol and hydroxy sulfate ester formation from the reaction of epoxides in tropospheric aerosols. Environ. Sci. Technol., 43(5), 1386–1392, doi: 10.1021/es8029076, 2009

Odum, J. R., Jungkamp, T., Griffin, R. J., Forstner, H., Flagan, R. C., and Seinfeld, J.H.: Aromatics, reformulated gasoline, and atmospheric organic aerosol formation, Environ. Sci. Technol., 31(7), 1890-1897, doi:10.1021/es960535l, 1997.

---

## Author Comment (AC2) · 5 May 2016

***Referee's Comment in Italic Font;*** **Author's response in Red** **and Manuscript revision in Blue without Italic Font**.

*The manuscript presents information on yields for the photooxidation of single ringed aromatic structures. The senior author has been measuring yields from aromatic hydrocarbons (AHCs) for more than 15 years and is well versed. In this particular manuscript, the structure of the aromatic hydrocarbon (AHC) has been varied to examine differences in the organic aerosol (OA) yield. Thus, a series of 12 alkyl-substituted C8 and C9 AHCs have been examined. For these experiment, the aerosol yield has been determined using the Odum two-product model. Other OA parameters examined include the ratio of aerosol mass spectrometry (AMS) peaks attributed to OA, the OAoxygen-to-carbon (O: C) ratio, the oxidation state (OS), density, and volatility. The authors conclude that changes in the OA chemical composition and volatility influences the yield typically by increasing the mass for increased oxidation. The authors also consider the atmospheric implications of this study.*

*(1)The study addresses an issue of perhaps abstruse importance. The oxidation of alkyl substituted AHCs and the formation of secondary organic aerosol (SOA) has been examined extensively over the last 20 years and this work appears to cover some old territory. Many of these topics were addressed in Odum et al. 1997a, b (authors' references) and the present manuscript provides a bit more insight.*

Odum's work was very important and provided a practical way to simplify aerosol yields. However, work over last decade has suggested the importance of $NO_x$ to SOA formation from aromatic hydrocarbons (e.g., Song, et al., 2005) with increasing aerosol formation observed for aromatics as initial NO levels are decreased. Lowering $NO_x$ conditions from the earlier Odum work improves representation of ambient conditions. Therefore, a comprehensive reinvestigation including isomer effects on SOA formation at more realistic hydrocarbon and NOx conditions is needed. Further, as noted by the reviewer, additional instrumentation available provides more insights into the SOA formation.

Higher yields are observed in this work for low $NO_x$ conditions than the earlier high $NO_x$ conditions (e.g Odum, et al, 1997, Figure 1, $M_0$=40, yield= ~0.03 or ~0.06; current work $M_0$=40, yield 0.07-0.12). This paper demonstrates the molecular structure impact of aromatic hydrocarbons on SOA formation including impacts on SOA yield, chemical composition and physical composition. This is the first comprehensive analysis of SOA formation from aromatic isomers since the original Odum work on SOA from aromatics. The previous Odum work provides only very limited experimental work on isomers which is insufficient to determine molecular impact on SOA formation. (e..g, p-xylene and o-xylenehave only two experiments)

*(2) Admittedly, the AMS was not around and the aerosol density from the volume distribution was considered to be unity in the 1990s. However, this work also represents a step backwards. Whereas Odum et al. 1996, 1997a sought to simplify aerosol yields, this work goes in the other*

*direction and makes an argument (at least implicitly) that the yields should be addressed more precisely, a contention that I don't feel has been justified (see Table 2).*

This work provides yield information with greater precision to dig into the role of molecular structure in SOA formation from aromatic hydrocarbon. We can't agree more with the reviewer that a simplified curve is a more attractive method for the curve fitting. Prior to further simplification, it is necessary to identify the relative importance of aromatic structure (o, m, p; alkyl length) especially when looking to project these findings for additional aromatic isomers. We currently have a paper under review to further demonstrate a novel method to simplify SOA yield from aromatic hydrocarbons, which requires insight on the relative importance of aromatic molecular structure. This work here focuses on molecular structure impact before stepping largely forward to the general trends found in SOA formation from aromatic hydrocarbons. The limited data sets available in earlier years are insufficient to reveal the difference among isomers and therefore might provide some bias on the similarity among aromatic hydrocarbon SOA formation, especially when conducted under high $NO_x$ conditions. As demonstrated in the manuscript, the difference among SOA from aromatic isomers, including SOA yield (for example, para position has significantly lower SOA yield compared with ortha and metal position), chemical composition and physical composition, does exist and should not be ignored by oversimplification. Therefore, this work is valuable to understand SOA formation from aromatic hydrocarbons before generalizing SOA aromatic yield trends.

*(3) That said, the experiments appear to have been carefully performed and there certainly are enough of them. Unfortunately, the initial conditions are all over the map and makes it very difficult to get a sense of the reproducibility of a given experiment. Virtually, nothing is said about uncertainty.*

There is only ~10% of experiments with HC/NO larger than 60 ppbC:ppb. The majority of experiments have similar NO conditions (see reply to referee #1 in Comment 4). Also, the similar impact of NO on radical and organic nitrate formation is demonstrated to be insignificant (see reply to referee #1 in Comment 4. The uncertainty of experiment is 6.6% based on ten repeat m-xylene experiments (please see referee #1 in Comment 3). Uncertainty in all analysis is included in the updated manuscript.

*1. My major substantive comments regarding the manuscript are as follows: (1) these experiments hardly qualify as being under low NOx conditions. The removal of RO2 radicals competitively by NO or RO2 determines the regime that the reaction is in. I would consider the low NOx regime as conditions where the RO2 radical-radical reactions become more important than the NO reaction. As a rule of thumb, I would say that this is certainly at no more than 5 ppb of NO for the conditions of these experiments depending, of course, on the specific RO2 radicals from the precursor AHC.*

(1) From a kinetics perspective, low $NO_x$ is even lower than the 5 ppb (suggested by the reviewer) tending to occur at NO levels in the 15 to 50 ppt range. However, starting with $NO_x$ levels < 50 ppt is substantially lower than practical experimental constraints for Teflon

environmental chambers due to offgasing of HONO from Teflon surfaces (Carter et al., 2005). These experiments are referred to as low $NO_x$ experiments to be consistent with environmental chamber literature over the last decade, which is referring to the relative amount of $NO_x$ at the beginning of the experiment when compared to initial VOC. The $NO_x$ ranges in this work are more consistent with urban $NO_x$ loadings than the earlier high $NO_x$ experiments performed by Odum (Odum, et al., 1997 and 1996) and others. Also, it is not possible to use an initial NO concentration ~5ppb since the low $NO_x$ concentration leads to a low reactivity of overall reaction and therefore forms less aerosol which is not atmospherically relevant.

Clearly the NO will compete for the $HO_2$ or $RO_2$ when there is sufficient NO. In other earlier work we demonstrate that SOA will not be formed until $NO_2/NO>70$ (Li, et al., 2015), which indicates that $RO_2$ majorly react with NO instead of $HO_2$ or $RO_2$ even at NO~5ppb under the range of NO we investigated. The important point is that the NO concentration is extremely low (<10ppt) during the majority of the photooxidation experiment (after onset of $O_3$ formation). $NO_x$ mainly exists as $NO_2$ when PM is formed and $HO_2+RO_2$ instead of NO+ $HO_2/RO_2$ dominates.

In order to clarify the $NO_x$ condition we actually used, we add the following information on Page 5 at Line 29 after "under low $NO_x$" add "(10-138 ppb)".

*(2)The version of SAPRC referenced to Carter and Heo (2013) is specifically geared to ozone prediction, that is, conditions where NOx dominates early product generations (two, at most three). As stated in Carter and Heo, (2013; Atmospheric Environment) SAPRC-11 is not geared for PM predictions. For this to be the case, the importance of RO2 + RO2 reactions should be adequately predicted as should the SOA mass. That is the point of the model and to predict radical concentrations as they were an end in themselves. The experimental SOA values could then constrain the model. That said, I question how well SAPRC-11 predicts radical concentrations under low NOx conditions. (3) Several sections need a complete writing overhaul. Section 3.2.1 is barely comprehensible. Many sections in the Results and Discussion presents data (e.g., S6) as if they were self-interpretive. The manuscript is written for the audience being other scientists in their research group. There is considerable jargon and the writing is highly imprecise hardly worthy of a scholarly journal. (4) To the extent possible, the authors should give a greater physical interpretation of the metrics they present in Section 3. Some are obvious (e.g., Sec. 3.2.2) others far less so (Sec. 3.2.1; 3.4).*

2) SAPRC-11 is geared to predict $O_3$ formation under low $NO_x$ conditions especially for aromatic hydrocarbons as described by Carter and Heo (2013). Literally, the $NO_x$ range we used in current work is within the used range of $NO_x$ when the model is updated to SAPRC-11. In fact, the aromatic experiments used to develop the SAPRC-11 update are included in this work. We agree that it could not well predict SOA formation since the gas phase products are not well demonstrated as suggest by Carter and Heo (2013), especially for those associated with gas to particle partitioning. However, SAPRC-11 should be sufficiently good to predict gas phase radical concentration, which is closely associated with ozone formation.

3) Section 3.2.1 extends the traditional $f_{44}$ vs $f_{43}$ ($C_2H_3O^+$) chemical composition analysis by including fragments ($C_3H_5O^+$ m/z 57 and $C_4H_7O^+$ m/z 71) from longer alkyl substitute other than methyl since longer alkyl substitutes are included in the isomers investigated. The goal of Section 3.2.1 is to provide insights into the SOA formation mechanism from different isomers as discussed in the later part of Section 3.2.1.

Some changes are already made in Section 3.2.1 (please see reply to referee #1, Comment 8). Also, we have added the following sentences on Page 11 at Line 3 to Section 3.2.1 .

This work extends the traditional $f_{44}$ vs $f_{43}$ ($C_2H_3O^+$) chemical composition analysis by including oxidized fragments ($C_3H_5O^+$ m/z 57 and $C_4H_7O^+$ m/z 71) of the longer (non-methyl) alkyl substitutes. Therefore, $f_{44}$ vs $f_{43}+f_{57}+f_{71}$ is plotted instead of $f_{44}$ vs $f_{43}$.

We demonstrate the calculation of H/C and O/C in section 2.2 Page 7 Line 16-19. We also described how the Figure S6 graph is made in the title of Figure S6. We add following sentences in the revised manuscript to better interpret Fig. S6 and other graphs (e.g. Fig. S3).

Further, we have added the following sentence on Page 7 Line 19:

Evolution of SOA composition (Heald, et al., 2010; Jimenez, et al., 2009) refers to the bulk SOA chemical composition changes with time. $f_{44}$ and $f_{43+57+71}$ evolution and H/C and O/C evolution refer to the change of $f_{44}$ and $f_{43+57+71}$ with time and the change of H/C and O/C with time, respectively.

Additionally, to address jargon concerns, we have had a couple of non-SOA focused experts in the air field review the paper to help identify and remove jargon along with the suggestions provided by the review.

4) The following revisions are made in Section 3.2.1 and Section 4 to improve the physical interpretation of the metrics used.

Sec. 3.2.1: The physical interpretation is improved after the revision described in 3) comment and the referee #1's Comment 8.   Sec. 4: The physical interpretation stated at the beginning of Section 4 as "Methyl dilution theory (Li, et al. 2015a) is extended to alkyl substitute dilution theory in order to investigate the influence of longer alkyl substitutes compared with methyl group substitutes." Additionally, the following sentence has been added to clarify the physical interpretation at Line 19 Page 17:

A robust prediction of SOA H/C and O/C trends for longer (C2+) alkyl substituted aromatics based on the methyl substituted aromatics will suggest a similarity in the role of methyl and longer alkyl to SOA formation; an underestimation or overestimation will indicate different oxidation pathways for aromatics with differing alkyl substitute length.

We also update the Fig 7 a & b according to referee #1's Comment 3 about standard deviation.

The implication from the difference between the measurement and prediction from the aromatics is updated correspondingly in the later part of Sect 4.

*Some comments and suggestions:*

2. *P5, L27. The goal of the research states the obvious. Perhaps more insight will motivate the reader to actually read the paper.*

We add the following sentence on Page 6 Line 2:

The effects of molecular structure impact on SOA yield, chemical composition (H/C, O/C, $OS_c$, $f_{44}$, $f_{43}$, $f_{57}$ and $f_{71}$) and physical properties (density and VFR) are demonstrated. Alkyl substitute dilution conjecture is further developed from methyl dilution theory (Li, et al., 2016).

3. *P6, L7. UV-350 bulbs have considerable radiation in the UVB which accentuates the photolysis of carbonyl compounds to a considerable extent which accelerates PM formation by increasing the radical concentrations. Thus, the two-product parameters developed (Table 2) may not be applicable for predictions of ambient AHC PM (Tables2 and S3)*

We agree that UV-350 bulbs do not provide the higher wavelength region which affects photolysis of certain carbonyl compounds. However, UV impacts on different carbonyl are different. The photolysis rate ratios with blacklights will be much lower in the chamber than in the atmosphere if carbonyls have action spectra similar to the α-dicarbonyls; however, blacklight photolysis rate ratios will be higher if carbonyls have action spectra more like that of acrolein (Carter, et al., 1995). The photolysis of carbonyl compounds are more likely to impact the radical concentration (e.g. OH) and may further impact the overall SOA formation by change the kinetic reactivity. Therefore, the light source impact on carbonyl photolysis turns out to be the influence of radical concentration on SOA formation. The difference in radical concentration between chamber and atmosphere is demonstrated in Li, et al., 2015.

Further work is needed to adjust the SOA yield concluded from current chamber studies to better predict the SOA formation under atmospheric conditions. The current work provides the fundamental data for further investigation. Therefore, we add to the paper the statement "Moreover, the five subcategories of aromatics and their two product modeling curve fitting parameters in this work at more realistic NOx loadings provide a more precise prediction of SOA formation form aromatic hydrocarbons under atmospheric conditions" in Section 5 for the atmospheric application as in referee #1 Comment 13. The current study is more focused on the isomer impact on SOA formation. The results and implications of the current study remain reasonable since all the precursors are studied under comparable conditions (see $k_{OH}$ discussion in the reply to referee #1 Comment 4 last part).

4. *P6, L18. Provide a chemical name for the standard OEKANAL.*

OEKANAL is a Sigma-Aldrich Grade (purity) for 1, 2, 3-trimethylbenzene and is not a

chemical name. It is followed by "1, 2, 3-trimethylbenzene Sigma-Aldrich".

5.  *P6, L23. How is mixing achieved in this large chamber; fan, diffusion, other?*

    The mixing prior to commencing an experiment is achieved by fans. See Carter, et al., 2005 for more details "The two reactors are connected to each other through a series of custom solenoid valves and blowers. The system provides for rapid air exchange prior to the start of an experiment ensuring, that both reactors have identical concentrations of starting material. Each reactor can be premixed prior to the start of an experiment by Teflon coated fans located within the reactor."   During the experiment the vibration on the chamber walls due to air circulation on the outside of the chamber provide sufficient mixing during the experiment.

6.  *P7, L10. Calling a bunch of peaks attributed to innumerable organic compounds completely fragmented by 70 eV electrons as a chemical composition stretches the concept of molecules beyond recognition. I would hardly call this metric a chemical composition in any traditional sense. Perhaps the word "effective" could be incorporated to indicate that this is simply a parameterized metric.*

    We agree that the chemical composition is derived from peaks from numerable organic compounds completely fragmented. There might be some difference in between the traditional definition and what is widely used nowadays to describe AMS chemical composition, which is a measure of the bulk chemical composition of the aerosol. We keep our manuscript consistent with recent publications using AMS results (eg., Crippa, et al., 2013; Lambe, et al., 2015).

7.  *P7, L23.The authors should reference the 2013 Atmospheric Environment article by Carter and Heo rather than the CARB report. The article went through peer-review and should be more reliable.*

    Done.

8.  *P8, L11. The sentence is unneeded; include the information in the caption.*

    Done.

9.  *P9, L3. To use the word "claimed" in a pejorative fashion is particularly bad form. If you believe the statement in Odum et al. is wrong, simply state it.*

    Fixed.   Changed "claimed" to "stated".

10. *P9, L14. Delete the sentence. The supplement does nothing to support the sentence other than to simply repeat itself and refer to a paper in preparation. Nothing is gain by including the sentence in the paper or in the supplement.*

    We keep this sentence to clarify the differences in the kinetics is insignificant (e.g., $k_{OH}$[OH],

[HO$_2$], …) and therefore the molecular structure of the isomers is driving the difference in SOA formation. The referenced paper is now published.

11. *P9, L23. Why is the assumption needed? The two-product model is just a fitting exercise anyway.*

Similar products are expected to be formed from the aromatic isomers. Fixing the K$_{om,2}$ value provides for similar treatment of the high volatility products allowing us to focus on the low volatility products most important to SOA formation under atmospheric conditions.

12. *P9. It might be worthwhile to examine partitioning using a volatility basis set (VBS) to see if any insight could be gained beyond the standard two-product fit which at this point is rather dated. This might provide a more useful metric for describing the partitioning of the AHC products.*

We agree that VBS is an attractive way to describe the SOA yield. However, VBS is fundamentally based on gas-particle partitioning theory which is the same as two-product model. The application of VBS only provide similar result in a different format. VBS presents the contribution of products with different volatility using bins and here two-product model use K$_{om}$. Therefore, we keep our analysis using the traditional two-product model.

13. *Sec 3.2.1. This section suffers from a lack of an understandable interpretation of the various fragments from the AMS output and their combinations into the combined metrics (e.g. Eq 1). An annoying aspect of this section is the comparison with other work before any interpretation is provided (e.g., P11, L5, 16; P12 L6, 25,. . ...). How do we even know that the conditions are applicable between these experiments and the ones being compared to?*

The interpretation is improved according to the reply to Comment 3). We cited other's work to provide the AMS result found in other chamber work for selected isomer species. The initial hydrocarbon and NO$_x$ conditions used in other's work are not completely the same as ours. Therefore, the AMS data is not exactly the same.    However, we demonstrate the AMS data we use are reasonably in-line with earlier studies to contextualize the results and demonstrate that further discussion of the AMS data is reasonable.

14. *P13, L7. LV-OOA and SV-OOA are presented both undefined and without context.*

We change "LV-OOA and SV-OOA" to "low volatility oxygenated organic aerosol (LV-OOA) and semi-volatile oxygenated organic aerosol (SV-OOA)". The definition of LV-OOA and SV-OOA can be found in detailed in Ng, et al 2011.

15. *P13, L8. The sentence as written belongs in the introduction. The intent of the sentence needs a rationale from the data. It is not self-evident.*

We added a sentence to the introduction part to emphasize this part as a reply to Comment 2 (P5, L27). The sentence referred to here is to transition from evolution data to average data. We

will delete it here and rewrite it as a sentence below in blue. We mention in the manuscript that "The evolution trend agrees with Fig. S3 (Sect. 3.2.1)." This means that the evolution trend is not significant during the photooxidation similar to what is mentioned in Section 3.2.1 for $f_{44}$ vs. $f_{43+57+71}$. Therefore, average value is sufficient to describe H/C and O/C. We add the following sentence in P13, L8 after "The evolution trend agrees with Fig. S3 (Sect. 3.2.1)." to provide the rationale.

..., which means no significant H/C and O/C evolution is observed in the current study. Therefore, average H/C and O/C with standard deviation provided is used to explore the impact of molecular structure on SOA chemical composition.

16. *P14, L21. "a more accurate metric. . .." More accurate than what?*

Fixed. Inserted "than H/C and O/C"

17. *P16, L1. Eliminate first name for Borrás.*

Fixed

18. *P17, L15. Use of the term "theory" (in any scientific sense) strikes me as somewhat pretentious. I would consider it as more of a conjecture.*

Good point. We use "Alkyl Dilution Conjecture" according to reviewer's suggestion; we keep "methyl dilution theory" as published in (Li, et al., 2016)

19. *P18, L8-16. What does the term "extremely low" mean? Provide a value for comparison. I would characterize most of this part as speculation. Also, experimental limitations in Forstner et al., 1997 (their specific quartz filter configuration) suggested that the furan-type compounds were in the particle phase but were almost certainly in the gas phase. The metrics in the present paper refer only to particle phase OA. The reference should be avoided here.*

We have now added the actual value for $OS_c$ and delete the reference in the revised manuscript. This part has been slightly modified after considering standard deviation/uncertainty as mentioned in response to Referee #1's Comment 3.

20. *Section 5. I seriously doubt that any of these parameterizations would appear in any wide-used air quality model. I would consider the work mainly for academic purposes.*

The SOA yield parameters are widely used in current model (e.g. CMAQ see Carlton, et al., 2010; GEOS-Chem see Heald, et al., 2011, WRF-CHEM model Li, et al., 2011 ). Current work provides improved SOA yield parameters than previous work under high-NOx conditions (e.g. Odum. et al., 1997).

21. *P19, L2. Replace "entend" with "extent".*

Fixed

22. *P19, L3. Nothing is "proved" here. The work simply provides "evidence for".*

Done

23. *P25, L8. Replace the ACPD manuscript with newly published ACP paper.*

Done

24. *Table 1, S2 and text. How many AHCs were studied: the text say twelve AHCs (P5,L7), Table 1 gives ten AHCs, and Tables 1 and S2 together give fourteen. Which is the right number? (For good measure, Figure 6 shows eleven precursors.)*

Twelve is the right number. Table 1 and Table S2 combine to give the 12 unique AHCs used in this study.

25. *Table 3. What's the point of the table if the p value are greater than 0.05. Certainly, the p-value for VFR and k (OH) is not zero.*

It is included in note below the table that "Alpha ($\alpha$) level used is 0.05. If the p-value of a test statistic is less than alpha, the null hypothesis is rejected". It the p value is greater than 0.05, the correlation found is not trusted within the $\alpha$ level. In another word, the larger the p-value the less confidence in the correlation provided. Certainly, the p-value for VFR and k (OH) is not absolute zero. However, it is <0.0005. Therefore, we change "0" into "0.000" for significant figure purposes.

26. *Figure 1. The final points control the shape of the curve. The data below 20 ug m-3 would be of most interest for atmospheric applications.*

The higher $M_0$ allows one to improve the parameters used to fit the overall aerosol formation trends, especially that for $\alpha_2$. The final points are therefore controlling the shape of the curve used to fit of the high volatility products, in this case $\alpha_2$. The identified curves reasonably represent the lower organic mass loadings (< 20 ug m$^{-3}$) as seen in the quality of the fit where $\alpha_1$ and $K_{om,1}$ (lower volatility products) dominate the shape of the curve. Curve fitting with and without $\alpha_2$ and $K_{om,2}$ are presented below.

[Figure]

27. *Figures 2, 3, 7. Am I missing something? Why are there no symbols for the values; is there that much uncertainty? The use of colors for the precursors in Table 2 is particularly annoying. Are not words sufficient without colors?*

The exact locations of these values show up after adding error bar to each value (see Referee #1 Comment 6's reply). I think you mean Figure 2 instead of Table 2. The colors we use categorize all the 12 isomers into different subgroups (e.g. all xylenes are in green as what we labeled in the upper right). We think these colors help the audience to understand what kind of molecular structure impact it is (location vs. length). We would like to keep the colors to help us demonstrate the findings.

28. *In Figure 4, what are the estimated uncertainties in the model-generate radical concentrations. Carter and Heo, 2013 suggests that these could be substantial.*

We are unsure of what the reviewer is requesting. Figure 4 provides "Oxidation state ($OS_c$) of SOA formed from different aromatic hydrocarbon" which nothing about model generated radical concentrations. Table S4 lists model-generated radical concentrations. Generally speaking, the [OH] is fitted through precursor measurement from GC-FID and therefore [OH] has little uncertainty (<~5%). SAPRC-11 adjusted photoreactive product quantum yield parameters are used to minimize average biases in Rate ($\Delta(O_3\text{-}NO)$) (Carter and Heo, 2013). The uncertainties of radical prediction is minimized since $O_3$ prediction relies on radical predictions. However, the uncertainties associated with SAPRC-11 is not a focus for current work. We provide the radical prediction provide by SAPRC-11 to rule out the impact of kinetic difference during the aromatic hydrocarbon photooxidation in order to emphasize the molecular structure impact.

29. *Table S3. How about the C9-trimethyl compounds studied? Why not put the parameters obtained from the TMB compounds in the table? It would also be informative to include the data for toluene (which must have been studied at some point) for comparison with the other single position substituents, ethylbenzene and n-,i-propylbenzene.*

Table S3 is used to support Fig. S1 to demonstrate the length effect among all $C_8$ and $C_9$. We don't study triple alkyl substitute that contain longer chains and therefore there is no need to list C9-trimethyl compounds in Table S3. (The fitting parameters for trimethylbenzenes can be found in Table 2.) We don't include toluene since it is not in the range of $C_8$ and $C_9$ aromatics; instead, we refer to our earlier work (Li, et al., 2016) at Page 10 Line 5.

30. *Table S4. The value of this table for predicting radical concentrations is very limited as noted above. It may be useful for urban NOx conditions but not where RO2 + RO2 is the dominant source of the aerosol. It doesn't surprise me that the p-values for virtually all comparisons in S5 are no different than the null hypothesis.*

We agree that SOA formation is tied to peroxide radical reactions. It should be noted that peroxide radical reaction is associated with NO, precursor concentration and other radicals (e.g. OH) as is ozone formation. We maintain that SAPRC could predict radical concentrations sufficiently well for how they are used in this paper (see response to comments above comment 1-2) and comment 28). The insignificant correlations between yield and radicals are not due to the limitation of the model but the similarity in kinetics among all the isomer precursor we studied. We actually found pretty good radical (eg. HO2/RO2) correlation with yield in our earlier work (Li, et al., 2015). Therefore, we prefer to keep the radical discussion as part of the supplement supporting the manuscript.

31. *SI Table 3 should be Table S6. Use lower case k in the table*

Fixed

**Reference**

Carter, W. P. L., Luo, D., Malkina, I. L. and Pierce, J. A.: Environmental Chamber Studies of Atmospheric Reactivities of Volatile Organic Compounds: Effects of Varying ROG Surrogate and NOx, Report to the California Air Resources Board Contracts (A032-0962), 1995

Carter, W. P. L., Cocker III, D. R., Fitz, D. R., Malkina, I.L., Bumiller, K., Sauer, C.G., Pisano, J.T., Bufalino, C., and Song, C.: A new environmental chamber for evaluation of gas-phase chemical mechanisms and secondary aerosol formation, Atmos. Environ., 39(40), 7768-7788, doi:10.1016/j.atmosenv.2005.08.040, 2005.

Carlton, A. G., Bhave, P. V., Napelenok, S. L., Edney, E. O., Sarwar, G., Pinder, R. W., Pouliot, G. A., and Houyoux, M.: Model representation of secondary organic aerosol in CMAQv4. 7, Environ. Sci. Technol., 44(22): 8553-8560, doi: 10.1021/es100636q, 2010.

Crippa, M., DeCarlo, P., Slowik, J. G., Mohr, C., Heringa, M. F., Chirico, R., Poulain, L.. Freutel, F., Sciare J., Cozic J., Di Marco, C.F.; Elsasser, M.; Nicolas, J.B.; Marchand, N.; Abidi, E.; Wiedensohler, A.; Drewnick, F.; Schneider, J.; Borrmann, S.; Nemitz, E.;

Zimmermann, R.; Jaffrezo, J.-L.; Prévôt, A.S.H., and Baltensperger, U.: Wintertime aerosol chemical composition and source apportionment of the organic fraction in the metropolitan area of Paris." Atmos. Chem. Phys., 13(2): 961-981, doi:10.5194/acp-13-961-2013, 2013.

Heald, C. L., Coe, H., Jimenez, J. L., Weber, R. J., Bahreini, R., Middlebrook, A. M., Russell, L. M., Jolleys, M., Fu T.-M., and Allan J. D.: Exploring the vertical profile of atmospheric organic aerosol: comparing 17 aircraft field campaigns with a global model, Atmos. Chem. Phys., 11(24): 12673-12696, doi:10.5194/acp-11-12673-201, 2011,

Jimenez, J. L., Canagaratna, M. R., Donahue, N. M., Prevot, A. S. H., Zhang, Q., Kroll, J. H., DeCarlo, P. F., Allan, J. D., Coe, H., Ng, N. L., Aiken, A. C., Docherty, K. S., Ulbrich, I. M., Grieshop, A. P., Robinson, A. L., Duplissy, J., Smith, J. D., Wilson, K. R., Lanz, V. A., Hueglin, C., Sun, Y. L., Tian, J., Laaksonen, A., Raatikainen, T., Rautiainen, J., Vaattovaara, P., Ehn, M., Kulmala, M., Tomlinson, J. M., Collins, D. R., Cubison, M. J., Dunlea1, E., J., Huffman, J. A., Onasch, T. B., Alfarra, M. R., Williams, P. I., Bower, K., Kondo, Y., Schneider, J., Drewnick, F., Borrmann, S., Weimer, S.,, Demerjian, K., Salcedo, D., Cottrell, L., Griffin, R., Takami, A., Miyoshi, T., Hatakeyama, S., Shimono, A., Sun, J. Y., Zhang, Y. M., Dzepina, K., Kimmel, J. R., Sueper, D., J. Jayne, T.,  Herndon, S. C., Trimborn, A. M., Williams, L. R., Wood, E. C., Middlebrook, A. M., Kolb C. E., Baltensperger, U. and Worsnop D. R.: Evolution of organic aerosols in the atmosphere, Science, 326(5959), 1525-1529, doi:10.1126/science.1180353, 2009.

Lambe, A. T., Chhabra, P. S., Onasch, T. B., Brune, W. H., Hunter, J. F., Kroll, J. H., Cummings, M. J., Brogan, J. F., Parmar, Y., Worsnop, D. R., Kolb, C. E., and Davidovits, P.: Effect of oxidant concentration, exposure time, and seed particles on secondary organic aerosol chemical composition and yield, Atmos. Chem. Phys., 15(6), 3063-3075, doi:10.5194/acp-15-3063-2015, 2015.

Li, G., Zavala, M., Lei, W., Tsimpidi, A., Karydis, V., Pandis, S. N., Canagaratna, M. and Molina L. : Simulations of organic aerosol concentrations in Mexico City using the WRF-CHEM model during the MCMA-2006/MILAGRO campaign, Atmos. Chem. Phys., 11(8): 3789-3809, doi:10.5194/acp-11-3789-2011, 2011,

Li, L.; Tang, P., Cocker III, D. R.: Instantaneous nitric oxide effect on secondary organic aerosol formation from m-xylene photooxidation. Atmos. Environ., 119, 144-155, doi:10.1016/j.atmosenv.2015.08.010, 2015.

Li, L., Tang, P., Nakao, S., Chen, C.-L., and Cocker III, D. R.: Role of methyl group number on SOA formation from aromatic hydrocarbons photooxidation under low-$NO_x$ conditions, Atmos. Chem. Phys., 16, 2255–2272, doi:10.5194/acp-16-2255-2016, 2016.

Odum, J. R., Jungkamp, T., Griffin, R. J., Forstner, H., Flagan, R. C., and Seinfeld, J.H.: Aromatics, reformulated gasoline, and atmospheric organic aerosol formation, Environ. Sci. Technol., 31(7), 1890-1897, doi:10.1021/es960535l, 1997.

Odum, J. R., Hoffmann, T., Bowman, F., Collins, D., Flagan, R. C., and Seinfeld, J. H.: Gas/particle partitioning and secondary organic aerosol yields, Environ. Sci. Technol., 30(8), 2580-2585, doi:10.1021/es950943+, 1996.

Song, C., Na, K., and Cocker III, D. R.: Impact of the hydrocarbon to $NO_x$ ratio on secondary organic aerosol formation, Environ. Sci. Technol., 39(9), 3143-3149, doi:10.1021/es0493244, 2005.

---

## Author Comment (AC3) · 5 May 2016

*Referee's Comment in Italic Font;* Author's response in Red and Manuscript revision in Blue without Italic Font.

*In this manuscript, the authors Li et al. studied formation of secondary organic aerosol from aromatic hydrocarbons, quantified the formation yields and investigated the chemical and physical properties in order to infer the differences in underlying mechanisms between different molecular structures. The hydrocarbons studied spanned from C7 to C9 and from 1 to 3 alkyl substitutes on the aromatic ring. These hydrocarbons are important for urban areas, and have not been well studied compared to benzene and toluene. The authors presented a variety of measurements (yields, aerosol mass spectrometer, density, volatility) conducted with state of the art instrumentation. The experiments are well designed and the manuscript is generally well written with relatively minor typos. However, I believe that the data interpretation is weak and it is very difficult to draw conclusions based on the data presented, especially with regards to SOA yields. I recommend major revisions before the manuscript can be considered for publication in ACP.*

*Major comments:*

*1. The biggest weakness of this manuscript is that the SOA yields are quite varied and many of the conclusions drawn by the authors are not very convincing. In Fig. 1, there are no error bars shown for the individual yields and Mo, and for the empirical fits. For example, the authors conclude from this figure that ortho compounds yields are similar and regression was performed on the combined data set. However, one could easily argue that o-xylene yields and m-ethyltoluene yields are similar and should be fitted to the same curve. Also, one of the major conclusions about ortho-compounds having higher yields is very weak. At atmospherically relevant Mo (Mo < 40 ug/m3), the yields are all very similar. Any differences in mechanisms do not translate to significant differences in formation yields (smaller than experimental uncertainties in yields).Therefore, one can argue that molecular structure plays a relatively minor role.*

We insert following sentence in method at Page 8 Line 9 to provide error bar for yields, as what replied to reviewer 1 in Comment 3: The uncertainty associated with 10 replicate *m*-xylene and NO experiments SOA yield is <6.65%.

We agree that yields are hard to distinguish by only looking limited datasets at a low mass loading (eg. $M_o$ < 40 ug/m$^3$). Our current fitting parameters already provide a good estimation for SOA formation under low mass loadings.

The argument about the minor role of molecular structure is not true as significant differences are suggested by the physical properties and chemical composition trend. The categorization of aromatic hydrocarbons for yield fitting are based on the measured data and the molecular structures such as the relative position of substitutes and the number of substitutes. Our yield categorization agrees with the observation found in later physical properties and chemical

composition trend part. For example, *o*-xylene is found to be more oxidized than m-ethyltoluene according to chemical composition (Fig. 2-4) and therefore it is reasonable to be categorized into different group.

*2. Another critique I have about this manuscript is the lack of mechanistic insights revealed by the experiments. I read the earlier paper on carbon dilution theory by the same authors, and found that the AMS data were very useful in identifying the general effect: methyl substituents divert OH oxidation from the ring, leading to less oxidized SOA components. A similar approach is used here but much less effectively. It is unclear from the proposed theory why the position of the alkyl substituents matters. Again, the authors claim that SOA from ortho-compounds is more oxidized than that from para- and meta-compounds. Why is that the case? The authors can make this manuscript much stronger if they can propose mechanisms along with careful experimental work to support them. Since the theme of the paper is molecular structure, I think these explanations are very important and deserve more attention. Can the authors propose any mechanisms (does not have to 100% proven, only needs to be plausible) that may be consistent with the results? AMS data are not suitable for resolving isomer-specific differences. More speciated measurements (such as LC or GC/MS) will provide more insights.*

We provide the mechanism leading to the difference in SOA formation from meta, para and ortho position in Section 3.2.2 Page 11, Line 20-Page 12 Line 10. Some mechanisms are also added as mentioned in the reply to Referee #1's Comment 12.

"A robust prediction of SOA H/C and O/C trends for longer (C2+) alkyl substituted aromatics based on the methyl substituted aromatics will suggest a similarity in the role of methyl and longer alkyl to SOA formation; an underestimation or overestimation will indicate different oxidation pathways for aromatics with differing alkyl substitute length." as described in the reply to Regeree #2's Comment 1-(4). The proposed theory is to demonstrate the impact of alkyl substitute length. The impact of position is filtered out since aromatics with similar alkyl location are used for the prediction as stated in Page 17 Line 21-24.

Following sentences is added on Page 19 Line 1:

"It is possible due to the alkyl substitute location impact on the further oxidation of five-membered bicyclic radicals. Different carbonyl compounds can form as the ring opening products from the dissociation of five-membered bicyclic radical. It is assumed that oligomerization of these carbonyl compounds can contribute to SOA (Li, et al., 2016). Aromatic hydrocarbons with para position alkyl substitute tend to form more ketone like dicarbonyl compounds than other aromatics. Ketone might contribute less to oligomerization formation compared with aldehyde as suggested in Li, et al (2016)".

The following sentence is added on Page 19 Line 2:

It might be due to a higher percentage of carbonyl with alkyl substitute formed during the oxidation of meta containing aromatics (e.g. methylgloxal, 2-methyl-4-oxopent-2-enal), which

contributes to oligomerization and thereby SOA formation.

More speciated measurements are warranted in future studies

*3. Similar to my previous comment (and other reviewers' comments), the AMS does not really provide information about specific chemical composition. For example, the statement in Section 3.2.2 "SOA components from all isomers are located in between slope= −1 and slope= −2 lines suggesting that SOA from these aromatic hydrocarbons is composed primarily of acid (carbonyl acid and hydroxycarbonyl) and carbonyl (ketone or aldehyde) like functional groups". The elemental ratios does not suggest that the SOA \*contain\* these functional groups, but rather they suggest that the SOA composition \*evolve\* as if they were adding these function groups. This is a very important difference that should be noted throughout the manuscript. The AMS elemental ratios help elucidate the bulk composition and evolution, not the specific chemical composition and mechanism, as suggested by the manuscript. I believe that AMS was useful in showing O/C changes from benzene to toluene to xylenes, but the specific isomeric differences in this paper are not convincingly argued in this manuscript.*

We agree that elemental ratios suggest that the SOA composition \*evolve\* as if they were adding these function groups. We will delete the following sentences:

"SOA components from all isomers are located in between slope=-1 and slope=-2 lines (Fig. S6) suggesting that SOA from these aromatic hydrocarbons is composed primarily of acid (carbonyl acid and hydroxycarbonyl) and carbonyl (ketone or aldehyde) like functional groups. The elemental ratio of SOA from p-xylene photooxidation was nearly located on the acid line (slope=-1)"

All the isomers start from the same precursor location in the Van Krevelen graph and therefore the difference in aerosol phase composition indicates the difference in oxidation. We agree that AMS is not specially for the detection of detailed species. However, the overall chemical composition provides the oxidation state of SOA and can be used to interpret the different extend of oxidation and related mechanisms.

*4. It is not clear to me why the authors chose to conduct the experiments under low NOx conditions. These highly substituted aromatic hydrocarbons are emitted in urban areas, and have lifetimes around 1 day or so. Even in rural areas, the NO levels are quite high, resulting in at least 50% oxidation by RO2+NO pathway (see Ortega et al., ACP, 2014). One could argue that the experiments conducted are still experiments with NOx (unlike HO2 dominated experiments). However, the HC/NOx ratios are not fixed (ranges over 1 order of magnitude), bringing into question the relative role of NOx. Why aren't the experiments conducted with a fixed HC/NOx ratio, or with NOx so high that it is not limiting?*

The low $NO_x$ we mentioned here is compared with earlier work as described on Page 4 Line 9. The $NO_x$ range we use is comparable to urban atmosphere. Range is clarified in revised manuscript (See reply to Referee #2 Comment 1-(1)) and is not the major driver of the

differences in SOA formation between the isomers. There might be a difference in the so called NO range when $RO_2+HO_2$ is dominated (See reply to Referee#2 Comment 1-(1)).

NO is depleted very fast at the beginning of the photooxidation and $HO_2$ and $RO_2$ reaction is dominating the major period of the photooxidation(see Reply to Referee #2 Comment 1-(1)).

*Other comments:*

*5. Pg 5 line 2: what is the significance of the furanone relative abundances? Does it point to a particular propensity of fragmentation and/or SOA formation from one particular molecular structure?*

The abundances of these products are

1.4 (± 0.39)% of 3-methyl-2,5-furanone is observed in toluene oxidation  (Forstner et al., 1997);

7.4 (±3.8) % 3-ethyl-2,5-furanone in ethylbenzene oxidation (Forstner et al., 1997).

The reference here is to give an example of molecular structure impact on aromatic oxidation products. The observed results in earlier work are not directly comparable to our work and therefore the exact abundance is not included in the manuscript.

*6. Pg 9 line 20: The parameters are for fitting purposes only, and are semi-empirical at best. Because the Odum 2-product equations are non-linear, alpha and K are coupled parameters (i.e. one can use a higher alpha and lower K and still get a decent fit to the experimental data). Therefore I suggest the authors not derive insights into relative volatilities from the fitted parameters.*

The semi-empirical two product model is based on the fundamental theory gas-particle partitioning. $\alpha_1$ and K have different impact on the curve and the pair we reported here is based on a best fit. It is partially empirical because two lumped groups are assumed. This model fitting parameters can't tell us which detailed species are favored in SOA from different aromatic. However, the difference among $\alpha$ and K provide the general information about the high volatility and low volatility products. Therefore, we would like to keep these implications from the yield parameters.

*7. Pg 14 line 2-11: Is it possible that photolysis of aromatic carbonyls is playing a role in affecting the H/C of SOA?*

It is possible. However, aldehydes higher than formaldehyde appear to react dominantly with OH radicals (Atkinson and Arey, 2003).

*8. Table 3: It seems to me that the correlations are quite weak (around 0.5 at best) and the p-*

*value is always greater than 0.05, sometimes much greater. From a statistics point of view, the correlations are inconclusive. I suggest rewording in main text to call these "weakly correlated". ("... best correlated..." is misleading).*

Done.

Use "correlated" instead of "best correlated". on Page16 Line 7

Delete "strong" before "correlation" on Page16 Line 25

Delete "well" before "correlated" on Page16 Line 2

*9. Sect 3.3.2: Can the authors isolate the effect of kinetics and molecular structure by comparing VFR at the same extent of reaction?*

All experiments are conducted under comparable kinetics and similar extent of reaction. (See $k_{OH}$ discussion in the reply to referee #1 Comment 4 last part).

*10. The authors alluded to the carbon dilution theory they proposed in an earlier publication (Li et al., 2015a) many times throughout the manuscript without explaining the theory. This theory is relatively novel and is not well known. Therefore I suggest adding a short section to explain the theory early on in the manuscript and show how this will be applied to the molecular structures studied in this work.*

Following sentence is added in revised manuscript on Page 17 Line 17 to add a short section to explain carbon dilution theory:

Carbon dilution theory proposed by Li et al (2016) successfully explain that methyl group impacts remain similar in SOA elemental ratios as in the aromatic precursor. The chemical composition of SOA formation from alkyl substituted aromatics is predicted by simply adding the alkyl substitute into the chemical composition of SOA formed from pure aromatic ring precursor (benzene).

We also add explanation in the revised manuscript to show how the theory is applied to this work. Please see reply to referee #2 1-(4)

*Technical comments:*

*11. Pg 2 Line 5: eight to nine carbon should be "C8- to C9-" or "eight- to nine-carbon"*

Done. Changed "eight to nine carbon" to "eight- to nine-carbon"

*12. Pg 2 Line 23 IPPC should be IPCC*

Fixed.

*13. Pg 2 Line 26 What does growth potential mean?*

Changed "have larger growth potential than biogenic aerosol sources" into "are more likely to increase"

*14. Pg 3 Line 14 Toluene and C8 aromatics do not dominate aromatic SOA, because they are not SOA themselves, but are precursors to SOA. I suggest rewording this sentence.*

Changed "Toluene and C8 aromatics dominate anthropogenic SOA" into "Toluene and $C_8$ aromatics dominate the anthropogenic SOA precursors".

*15. Pg 5 line 12: references to mass loading (Shilling et al. 2009 and Pfaffenberger et al 2013) and NO effect (Eddingsaas et al. 2012) on SOA formation are for a-pinene, not for aromatics. The authors should clarify that point.*

Add "(a-pinene)" after literature in revised manuscript to clarify the SOA precursor.

*16. Pg 6 line 13-20: are the chemicals used without further purification?*

Yes.

*17. Pg 7 line 8-9: "Volume fraction remaining" should be all capitalized*

Done.

*18. Pg 7 line 13-14: clarify that fx is the \*mass\* fraction of organic signal at m/z = x*

Done.

*19. Pg 7 line 19: "squirrel" should be capitalized*

Done. Change to "Squirrel 1.56D / Pika 1.15D"

*20.Pg 8 line 10: organic mass concentration should be Mo (o for organics), not M0. M0 would suggest that it is an initial mass concentration*

Fixed.

*21. Pg 10 line 17: what does the n stand for? It might be more useful to use n to denote the carbon number of the alkyl substitute. So for m/z 57 is derived from an ethyl-substituted aromatic, so n would be 2, and the formula would be C_(n+1)H_(2n+1)O+.*

Add "n =carbon number of the alkyl substitute" after $C_nH_{2n-1}O^+$

The formula should be $C_nH_{2n-1}O^+$ while using n to denote the carbon number of the alkyl substitute.

*21. Pg 10 line 21: "hydrogen-like organic aerosol" should be "hydrocarbon-like organic aerosol"; it seems quite obvious to me that it is not possible to have a C3H7 or C4H9 fragment from aromatic compounds here. Even isopropyl benzene does not yield C3H7 upon EI fragmentation (see NIST spectra). I suggest removing that explanation to make it more concise.*

Fixed. "hydrogen-like organic aerosol" should be "hydrocarbon-like organic aerosol"

We want to keep the explanation to clarify the different between chamber conditions and ambient.

*22. Pg 11 line 25: This section is very hard to follow. Is there a proposed mechanism of how an oxidation product of aromatic compounds can produce m/z 43 that is also consistent with proposed carbon dilution theory? It would be very beneficial here to use a figure to illustrate the key points.*

We add Figure in supplemental materials to demonstrate the proposed mechanism. (New Fig. S7)

*23. Pg 12 line 1: do the authors mean ROOH here? The bicyclic peroxides formed in aromatic oxidation are internally bridged, and are therefore ROOR, not ROOH (or hydroperoxides). I presume that in the presence of NOx, RO2+HO2 is negligible and ROOH is not formed. Also, is there a reference for peroxides not yielding CO2+ from the AMS literature?*

Change "bicyclic hydroperoxides" to "bicyclic peroxides" $RO_2+HO_2$ dominated since ozone is formed (see Reply to Comments 4).

$CO_2^+$ can come from carbonates, cyclic anhydrides and lactones (McLafferty and Turecek, 1993). This indicates that the $CO_2^+$ should come from a fragments with -O-C-O- structure. Neutral $CO_2$ should be formed before EI to generate $CO_2^+$. Thermal decarboxylation is a possible pathway to form $CO_2$ from compounds such as aliphatic acid (McLafferty and Turecek, 1993). We hypothesize that it is impossible to form a neutral $CO_2$ if $CO_2^+$ comes from -C-O-O-. We do not find a reference to support that peroxides not yielding $CO_2^+$ from the AMS.

We deleted "More importantly, the difference in f44 implies that substitute location influences the further reaction pathway to form $CO_2^+$ since $CO_2^+$ is not readily available from bicyclic hydroperoxides." Add following sentences to demonstrate the source of $CO_2^+$ on page 11 Line 26.

$CO_2^+$ are generally formed during MS electrical ionization from carbonates, cyclic anhydrides

and lactones (McLafferty and Turecek, 1993) indicating that the $CO_2^+$ is associated with -O-C-O- structure. Within the AMS, the $CO_2^+$ is also associated with decarboxylation of organic acids during heating followed by electrical ionization of the $CO_2$. We hypothesize that $CO_2^+$ formation from bicyclic peroxides is insignificant since $CO_2$ loss is not expected to come from -C-O-O- structure during thermal decomposition. Therefore, it is the reaction products of bicyclic peroxides that lead to the formation of $CO_2^+$ and the difference in $f_{44}$.

*24. Pg 13 line 8: "The current study concentrated on" should be "The current study concentrates on"*

Fixed.

*25. Pg 17 line 20: the chemical formula C2H2n+1 seems wrong*

Fixed. It should be $C_nH_{2n+1}$

*26. Figure 2: there should not be a continuous scale for molecular structure; Also for all the figures, the experiment numbers should be noted as such. Otherwise the numbers do not mean anything to readers*

Used color legend instead of color scale in revised manuscript.

*27. Figure 4: can the bars be color coded to correspond to those in other figures (e.g. Figs.5 and 6)?*

Done

---

## Referee Report (RR1)

This manuscript by Li et al. describes observations of structural impact on SOA formation from C8-C9 aromatic hydrocarbons in chamber studies. This work provides comprehensive dataset of SOA formation, including SOA yield data, elemental chemical composition data from AMS, and SOA density and volatility data. Although SOA formation from aromatic hydrocarbons has been studied for over 20 years (as the authors cited), the insights from the analyses using new techniques and new knowledge are beyond the level that scientists had discovered 20 years ago. Thus I think this manuscript is appropriate to be published in ACP. In the revised manuscript, the authors have addressed most of the comments from previous reviewers, but some small issues still need to be addressed, mostly related to the SOA yield part:

1. It might be misleading to use the term "low NOx" to describe the experimental conditions. 20-140 ppb of NO is relatively "low NOx" in chamber studies, but not the same case for ambient conditions. This point was also raised by Reviewer #2. I think what really matters is not the NOx level, but the relative branching ratios of different RO2 reaction pathways (RO2 + NO vs. RO2 + HO2 vs. RO2 + RO2), which can be simulated/estimated using SAPRC. For example, in experiment 1226B and 1421A, with very distinct NO levels, if the authors provide a simulation result (in main paper or supplemental information) to show that under such a range of NO levels, RO2 + HO2 is always the dominant pathway (e.g., >70% in these experiments), then I think the argument about the NOx level is clarified.

2. It is also useful to provide AMS mass spectra comparison of SOA from the same aromatic hydrocarbon under different HC:NOx conditions. How much change occurred to SOA chemical composition when initial HC:NOx condition changed? Without clarification of these changes, it is difficult to argue that the SOA yield difference is mainly due to different molecular structure, not different products under various experimental conditions.

3. The authors claimed that ortho position substituted aromatic hydrocarbons have highest SOA yield, based on observations. I think the authors should add some more discussion and provide some mechanistic insights to explain these observations, combining with the AMS chemical composition data. The discussion in Section 5, from line 14 is helpful to understand the observation, but need to re-organize and extend to a separate section with chemical structure schemes, if possible. For example, are ortho position substituted aromatic hydrocarbon products less likely to fragment due to the structure?

---

## Author Response (AR2)

***Referee's Comment in Italic Font;*** **Author's response in Red** **and Manuscript revision in Blue without Italic Font**.

*This manuscript by Li et al. describes observations of structural impact on SOA formation from C8-C9 aromatic hydrocarbons in chamber studies. This work provides comprehensive dataset of SOA formation, including SOA yield data, elemental chemical composition data from AMS, and SOA density and volatility data. Although SOA formation from aromatic hydrocarbons has been studied for over 20 years (as the authors cited), the insights from the analyses using new techniques and new knowledge are beyond the level that scientists had discovered 20 years ago. Thus I think this manuscript is appropriate to be published in ACP. In the revised manuscript, the authors have addressed most of the comments from previous reviewers, but some small issues still need to be addressed, mostly related to the SOA yield part:*

*1. It might be misleading to use the term "low NOx" to describe the experimental conditions. 20-140 ppb of NO is relatively "low NOx" in chamber studies, but not the same case for ambient conditions. This point was also raised by Reviewer #2. I think what really matters is not the NOx level, but the relative branching ratios of different RO2 reaction pathways (RO2 + NO vs. RO2 + HO2 vs. RO2 + RO2), which can be simulated/estimated using SAPRC. For example, in experiment 1226B and 1421A, with very distinct NO levels, if the authors provide a simulation result (in main paper or supplemental information) to show that under such a range of NO levels, RO2 + HO2 is always the dominant pathway (e.g., >70% in these experiments), then I think the argument about the NOx level is clarified.*

It is a good suggestion. The following graphs are added into supporting information as Fig. S1 to show that under such a range of NO levels, $RO_2 + HO_2$ is always the dominant pathway (e.g., >70% in these experiments)

[Figure]

[Figure]

*2. It is also useful to provide AMS mass spectra comparison of SOA from the same aromatic hydrocarbon under different HC:NOx conditions. How much change occurred to SOA chemical composition when initial HC:NOx condition changed? Without clarification of these changes, it is difficult to argue that the SOA yield difference is mainly due to different molecular structure, not different products under various experimental conditions.*

The average AMS mass spectra of SOA formation from *o*-xylene under different initial NO [50 ppb NO+ 80 ppb o-xylene (1320A) and 20ppb NO + 80ppb *o*-xylene (repeat experiment of 1321A) is provided in the graph below. We also compare the difference of SOA formed under different NO conditions in $f_{44}$ vs $f_{43}$ and H/C vs O/C graphs. There is no significant chemical composition difference observed in SOA formed under different NO conditions.

[Figure]

[Figure]

*3. The authors claimed that ortho position substituted aromatic hydrocarbons have highest SOA yield, based on observations. I think the authors should add some more discussion and provide some mechanistic insights to explain these observations, combining with the AMS chemical composition data. The discussion in Section 5, from line 14 is helpful to understand the observation, but need to re-organize and extend to a separate section with chemical structure schemes, if possible. For example, are ortho position substituted aromatic hydrocarbon products less likely to fragment due to the structure?*

Thanks reviewer for the suggestion. We discuss about the promoting of SOA formation by the ortho position in Section from line 14 to Line 25 by suggesting the different mechanism associated with different molecular structure (ortho, para and meta). We would like to add more discussion on that if we could. However, our study is based on the bulk chemical composition measured on AMS. We do not have enough information to make a justifiable discussion. Further study on the speciation of SOA components is suggested to improve the understanding of SOA formation mechanism from different isomer.

The Figure numbers used in the manuscript are modified accordingly since a new Fig. S1 is added as a response to the first comment of reviewer #4.

[revised manuscript text omitted]

* Experimental runs same as Fig. S4.

[Figure]

(a)

(b)

Fig. S1 Time series concentration change of NO, HO$_2$· and RO$_2$· during the phototoxidation of aromatic hydrocarbons. (m-Ethyltoluene, 1226B and 1421A; radical concentrations are predicted by SAPRC-11)

[Figure]

Fig.S1S2. Aromatic SOA yields as a function of $M_0$ for single compounds: a) Ethylbenzene (dotted line) vs Propylbenzene (i- and n-)(dashed line); b) *o*-Xylene(dotted line) vs *o*-Ethyltoluene(dashed line); c) *m*-Xylene(dotted line) vs *m*-Ethyltoluene(dashed line); d) *p*-Xylene (dotted line) vs *p*-Ethyltoluene (dashed line) (Note: Song, et al, 2005; Song, et al, 2007; Li, et al., 2016 data are also included)

[Figure]

Fig.  S3 Relationship between SOA yield and HO$_2$· radical concentration (Colored with substitute number or position and sized with mass loading M$_0$)

[Figure]

[Figure]

Fig.  S4  $f_{44}$ and $f_{43+57+71}$ evolution in SOA formed from different aromatic hydrocarbon photooxidation under low $NO_x$; each marker type represents one aromatic hydrocarbon and marker is colored with photooxidation time from light to dark: a) Ethylbenzene 2048A; b) Propylbenzene 1245A; c) Isopropylbenzene 1247A; d) $m$-Xylene 1191A; e) $m$-Ethyltoluene 1199A; f) $o$-Xylene 1320A; g) $o$-Ethyltoluene 1179A; h) $p$-Xylene 1308A; i) $p$-Ethyltoluene 1194A; j) 1,2,3-Trimethylbenzene 1162A; k) 1,2,4-Trimethylbenzene 1119A; l) 1,3,5-Trimethylbenzene 1156A

[Figure]

[Figure]

b-4)

c-1)

c-2)

c-3)

c-4)

[Figure]

d-1)

d-2)

e-1)

e-2)

e-3)

[Figure]

f-1)

f-2)

g-1)

g-2)

g-3)

[Figure]

h-1)

h-2)

i-1)

i-2)

i-3)

[Figure]

[Figure]

l-1)

l-2)

l-3)

Fig.  S5   High-resolution mass spectra of m/z 43, m/z 44, m/z 57 and m/z 71 measured for secondary organic aerosol formed at peak aerosol concentration during aromatic photooxidation a) Ethylbenzene 1146A; b) Propylbenzene 1245A; c) Isopropylbenzene 1247A; d) *m*-Xylene 1191A; e) *m*-Ethyltoluene 1199A; f) *o*-Xylene 1320A; g) *o*-Ethyltoluene 1179A; h) *p*-Xylene 1308A; i) *p*-Ethyltoluene 1194A; j) 1,2,3-Trimethylbenzene 1162A; k) 1,2,4-Trimethylbenzene 1119A; l) 1,3,5-Trimethylbenzene 1156A (m/z 57 only includes $C_3H_5O^+$). *m/z peaks with intensity less than 0.1 are not displayed

[Figure]

Fig.  S6 m/z 43, m/z 44, m/z 57 and m/z 71 fraction in SOA formed from aromatic
hydrocarbon photooxidation

* Oxidation products of the bicyclic radicals lead to the formation of $C_2H_3O^+$. The bicyclic radicals show the origin of $C_2H_3O^+$ during the oxidation of aromatic hydrocarbons.

Fig. S6 S7 The potential relationship between alkyl substitute location and the $C_2H_3O^+$ fragments from HR-TOF-AMS.

[Figure]

[Figure]

Fig.  S8 H/C and O/C evolution in SOA formed from different aromatic hydrocarbon photooxidation under low NO$_x$; each marker type represents one aromatic hydrocarbon and marker is colored with photooxidation time from light to dark: a) Ethylbenzene 1146A; b) Propylbenzene 1245A; c) Isopropylbenzene 1247A; d) *m*-Xylene 1191A; e) *m*-Ethyltoluene 1199A; f) *o*-Xylene 1320A; g) *o*-Ethyltoluene 1179A; h) *p*-Xylene 1308A; i) *p*-Ethyltoluene 1194A; j) 1, 2, 3-Trimethylbenzene 1162A; k) 1, 2, 4-Trimethylbenzene 1119A; l) 1, 3, 5-Trimethylbenzene 1156A.

[Figure]

Fig.  S9 H/C vs. O/C in SOA formed from different aromatic hydrocarbon photooxidation under low NO$_x$ colored by aromatic isomer type and marked with individual aromatic hydrocarbon species (C8 and C9 on the lower left indicate the location of initial aromatic hydrocarbon precursor; dashed/solid line indicate that changes between precursor and SOA components): Ethylbenzene 2084A; Propylbenzene 1245A; Isopropylbenzene 1247A; *m*-Xylene 1191A; *m*-Ethyltoluene 1199A; *o*-Xylene 1320A; *o*-Ethyltoluene 1179A; *p*-Xylene 1308A; *p*-Ethyltoluene 1194A; 1, 2, 3-Trimethylbenzene (123TMB) 1162A;1, 2, 4-Trimethylbenzene(124TMB) 1119A; 1, 3, 5-Trimethylbenzene(135TMB) 1156A. Alkyl number trend is the liner fitting in (Li., et al., 2015a).

[Figure]

Fig S9 S10 Relationship between VFR and $K_{OH}$ during aromatic hydrocarbon photooxidation under low NO$_x$